# Projected Randomized Smoothing for Certified Adversarial Robustness

**Samuel Pfrommer**                                                *sam.pfrommer@berkeley.edu*
*Department of Electrical Engineering and Computer Sciences*
*University of California, Berkeley*

**Brendon G. Anderson**                                            *bganderson@berkeley.edu*
*Department of Mechanical Engineering*
*University of California, Berkeley*

**Somayeh Sojoudi**                                                *sojoudi@berkeley.edu*
*Department of Electrical Engineering and Computer Sciences*
*Department of Mechanical Engineering*
*University of California, Berkeley*

**Reviewed on OpenReview:** *https://openreview.net/forum?id=FObkvLwNSo*

## Abstract

Randomized smoothing is the current state-of-the-art method for producing provably robust classifiers. While randomized smoothing typically yields robust $\ell_2$-ball certificates, recent research has generalized provable robustness to different norm balls as well as anisotropic regions. This work considers a classifier architecture that first projects onto a low-dimensional approximation of the data manifold and then applies a standard classifier. By performing randomized smoothing in the low-dimensional projected space, we characterize the certified region of our smoothed composite classifier back in the high-dimensional input space and prove a tractable lower bound on its volume. We show experimentally on CIFAR-10 and SVHN that classifiers without the initial projection are vulnerable to perturbations that are normal to the data manifold and yet are captured by the certified regions of our method. We compare the volume of our certified regions against various baselines and show that our method improves on the state-of-the-art by many orders of magnitude. [1].

## 1 Introduction

Despite their state-of-the-art performance on a variety of machine learning tasks, neural networks are vulnerable to adversarial inputs—inputs with small (often human-imperceptible) noise that is maliciously crafted to induce failure (Biggio et al., 2013; Szegedy et al., 2014; Nguyen et al., 2015). This sensitive behavior is unacceptable in contemporary safety-critical applications of neural networks, such as autonomous driving (Bojarski et al., 2016; Wu et al., 2017) and the operations of power systems (Kong et al., 2017). The works Eykholt et al. (2018) and Liu et al. (2019a) highlight the validity and eminence of these threats, wherein both physical and digital adversarial perturbations are shown to cause image classification models to misclassify vehicle traffic signs.

Heuristics have been proposed to defend against various adversarial attacks, only to be defeated by stronger attack methods, leading to an "arms race" in the literature (Carlini & Wagner, 2017; Kurakin et al., 2017; Athalye et al., 2018; Uesato et al., 2018; Madry et al., 2018). This has motivated researchers to consider certifiable robustness—theoretical proof that models perform reliably when subject to arbitrary attacks of a

---

[1]Source code for reproducing our results is available on GitHub.

bounded norm (Wong & Kolter, 2018; Weng et al., 2018; Raghunathan et al., 2018; Anderson et al., 2020; Ma & Sojoudi, 2021). Randomized smoothing, popularized in Lecuyer et al. (2019); Li et al. (2019); Cohen et al. (2019), remains one of the state-of-the-art methods for generating classifiers with certified robustness guarantees. Instead of directly classifying a given input, randomized smoothing intentionally corrupts the input with random noise and returns the most probable class, which, intuitively, "averages out" any potential adversarial perturbations in the data.

The seminal work Cohen et al. (2019) certifies that no adversarial perturbation within a certain $\ell_2$-ball can cause the misclassification of a smoothed model using isotropic Gaussian noise of a fixed variance. Recent works have attempted to certify larger regions of the input space by turning to randomized smoothing with optimized variances (Zhai et al., 2020), input-dependent variances (Alfarra et al., 2020; Wang et al., 2021), anisotropic distributions (Eiras et al., 2021), and semi-infinite linear programming (Anderson et al., 2022). However, for a fixed variance, the certified radius is upper-bounded by a constant in the dimension $d$ of the input (Kumar et al., 2020), implying that the volume of the certified $\ell_2$-ball degrades factorially fast as $O(K^d \Gamma(\frac{d}{2}+1)^{-1})$, where $\Gamma$ is Euler's gamma function and $K$ is some positive constant (Folland, 1999). Current input-dependent and anisotropic smoothing approaches have similarly been shown to suffer from the curse of dimensionality (Súkeník et al., 2021).

The small certified regions of randomized smoothing in high dimensions corroborate empirical findings that show increased robustness when precomposing classifiers with dimensionality reduction, e.g., principal component analysis projections (Bhagoji et al., 2018) and autoencoders (Sahay et al., 2019). These findings align with the manifold hypothesis, which posits that real datasets lie on a low-dimensional manifold in a high-dimensional feature space (Fefferman et al., 2016), and related results showing that perturbation directions most useful to an adversary are ones normal to this manifold (Jha et al., 2018; Zhang et al., 2020b). Thus, projecting inputs onto the manifold, or at least a low-dimensional subspace containing the manifold, should increase classification robustness. Methods taking this approach, such as Mustafa et al. (2019) and Alemany & Pissinou (2022), have worked well as heuristics, but lack theoretical robustness guarantees. Motivated by these works, we aim to enlarge the certifiably robust regions of randomized smoothing by performing the smoothing in a low-dimensional space in which adversarial access to the data's statistically insignificant yet vulnerable features has been eliminated.

## 1.1 Contributions

We propose *projected randomized smoothing*, whereby inputs are projected onto a low-dimensional linear subspace in which randomized smoothing is applied before classification. Our method combines the empirical successes of dimension-reducing projection methods with the theoretical guarantees of randomized smoothing to achieve the following contributions:

1. We theoretically characterize the geometry of the certified region in the input space and prove a tractable lower bound on the volume of this certified region.

2. We empirically demonstrate that classifiers can be attacked along subspaces spanned by statistically insignificant features that contribute nothing to classification accuracy, which are vulnerabilities that projected randomized smoothing certifiably eliminates.

3. Experiments on CIFAR-10 (Krizhevsky et al., 2009) and SVHN (Netzer et al., 2011) show that our method yields certified regions with order-of-magnitude larger volumes than prior smoothing schemes.

## 1.2 Related works

**Robustification via dimensionality reduction.** The work Bhagoji et al. (2018) was the first to consider linearly projecting inputs onto the top principal components of the training data before classification as a means to improve empirical (not certified) robustness. The authors of Sahay et al. (2019) nonlinearly preprocess test data using denoising and dimension-reducing autoencoders, and find a substantial increase in

classification accuracy when the inputs are subject to the popular fast gradient sign method attack. The work Bafna et al. (2018) projects an input onto its top-$k$ discrete cosine transform components to defend against "$\ell_0$"-attacks, but this empirical defense was later broken using adapative "$\ell_0$"-attacks (Tramèr et al., 2020), which directly motivates our approach for certified projection-based robustness. The work Sanyal et al. (2018) introduces a low-rank regularizer to encourage neural network feature representations to reside in a low-dimensional linear subspace, which is found to enhance empirical robustness. In Mustafa et al. (2019), the authors use super-resolution to project images onto the natural data manifold and obtain high empirical robustness for convolutional neural networks. Alemany & Pissinou (2022) shows that decreasing the codimension of data, i.e., decreasing the difference between the intrinsic dimension of the data manifold and the dimension of the input space in which it is embedded, generally leads to increased robustness of models defined on that input space.

Shamir et al. (2021) posits that learned decision boundaries tend to align with and "dimple" around the natural data manifold, and that adversarial perturbations are normal to this manifold. This finding supports our approach for certifiably eliminating off-manifold perturbations by projecting onto a low-dimensional approximation of the data manifold. The authors of Awasthi et al. (2021) reformulate principal component analysis to find projections that are robust with respect to projection error—a method that naturally complements our framework—and give robustness guarantees for the Bayes optimal projection-based classifier in the special case of binary Gaussian-distributed data. The work Zeng et al. (2021) precomposes classifiers with orthogonal encoders and performs randomized smoothing in the encoder's low-dimensional latent space as a means to speed up the sample-based smoothing procedure. To the best of our knowledge, Zeng et al. (2021) is the only work that provides certified robustness guarantees for general models and data distributions when using dimensionality reduction at the input—all of the other referenced works are heuristic—and their choice of orthogonal encoders ensures that the certified $\ell_2$-ball in the input space has the same radius as that in the latent space. Notably, their approach is highly conservative in estimating the input-space certified set as it relies on Lipschitzness of the orthogonal encoding layers, and is thus employed primarily as a means to speed up randomized smoothing. On the other hand, the method we propose uses a robustification-motivated projection for which we prove more general (anisotropic) certicates that capture off-manifold perturbations.

**Certification via randomized smoothing.** The work Cohen et al. (2019) develops randomized smoothing using an isotropic Gaussian distribution with input-independent variance to obtain certified $\ell_2$-balls. A subsequent line of works attempts to generalize randomized smoothing to other classes of certified regions, e.g., Wasserstein, "$\ell_0$"-, $\ell_1$-, and $\ell_\infty$-balls (Levine & Feizi, 2020; Lee et al., 2019; Teng et al., 2020; Yang et al., 2020). Various approaches have been taken to enlarge the certified regions. For example, Salman et al. (2019) unifies adversarial training with randomized smoothing to obtain state-of-the-art certified $\ell_2$-radii. The authors of Zhai et al. (2020) incorporate the certified $\ell_2$-radius into the model's training objective as a means to enlarge certified regions. The method in Zhang et al. (2020a) optimizes over base classifiers to increase the size of more general $\ell_p$-balls. Li et al. (2022) employs a second smoothing distribution to tighten robustness certificates.

Optimizing the certified region pointwise in the input space has also been considered, but generally these methods require locally constant smoothing distributions to ensure that the resulting certificates are mathematically valid (Alfarra et al., 2020; Wang et al., 2021; Súkeník et al., 2021; Anderson & Sojoudi, 2022). To further strengthen the robustness guarantees of randomized smoothing, the recent works Eiras et al. (2021); Erdemir et al. (2021); Tecot (2021) have turned to certifying anisotropic regions of the input space. For example, Eiras et al. (2021) maximizes the volume of certified ellipsoids and generalized cross-polytopes of the form $\{x \in \mathbb{R}^d : \|Ax\|_p \leq b\}$ for $p \in \{1, 2\}$, allowing for the certification of perturbations that are potentially larger in magnitude than the minimum adversarial perturbation. We show in Section 4 that our proposed method is able to outperform these methods by leveraging dimensionality reduction. As is standard practice in the randomized smoothing literature (Cohen et al., 2019; Yang et al., 2020; Jeong et al., 2021; Zhai et al., 2020; Lee et al., 2019), our emphasis is on certified robustness and not empirical robustness—we refer the reader to Maho et al. (2022) for connections between certified and empirical robustness under randomized smoothing, and in particular the difficulty in constructing and evaluating suitable empirical attacks.

We also emphasize that volume (Lebesgue measure) is the natural scalar measure for the size of anisotropic certified regions of the input space and is the standard notion considered by prior works (Liu et al., 2019b; Eiras et al., 2021; Tecot, 2021).

### 1.3 Notation

We denote the set of real numbers by $\mathbb{R}$. The $\ell_2$-norm of a vector $x \in \mathbb{R}^n$ is denoted by $\|x\|$, whereas the general $\ell_p$-norm is given an explicit subscript $\|x\|_p$. The range and nullspace of a matrix $U \in \mathbb{R}^{m \times n}$ are denoted by $\mathcal{R}(U) \subseteq \mathbb{R}^m$ and $\mathcal{N}(U) \subseteq \mathbb{R}^n$, respectively. The $n \times n$ identity matrix is written as $I_n$. For a random variable $X$ with distribution $\mathcal{D}$ and a measurable function $f$, the expectation of $f(X)$ is denoted by $\mathbb{E}_{X \sim \mathcal{D}} f(X)$. The multivariate normal distribution with mean $\mu \in \mathbb{R}^n$ and covariance $\Sigma \in \mathbb{R}^{n \times n}$ is given by $N(\mu, \Sigma)$. The cardinality of a set $S$ is written as $|S|$. For a Lebesgue-measurable set $S \subseteq \mathbb{R}^n$ contained in a $k$-dimensional affine subspace, we write $V_k(S)$, termed the $k$-dimensional volume of $S$, to mean the Lebesgue measure of $S$ within that affine subspace. For sets $S, T \subseteq \mathbb{R}^n$, we denote their Minkowski sum by $S + T = \{x + y : x \in S, \ y \in T\}$. Euler's gamma function is denoted by $\Gamma$. Recall that $\Gamma(n) = (n-1)!$ when $n$ is a positive integer.

## 2 Classifier architecture

Consider the task of classifying inputs from a zero-centered cube $C^d = [-1/2, 1/2]^d \subseteq \mathbb{R}^d$ into $c$ distinct classes $\mathcal{Y} = \{1, 2, \ldots, c\}$.[2] Under the randomized smoothing framework, we begin with a given classifier $f_\theta \colon \mathbb{R}^d \to [0, 1]^c$, parameterized by $\theta$, that maps into the probability simplex over $c$ classes. The problem at hand is to increase the robustness of $f_\theta$ with certifiable guarantees.

**Vanilla randomized smoothing.** We give a brief overview of how this would be accomplished using vanilla randomized smoothing (Cohen et al., 2019). Randomized smoothing takes the *base classifier* $f_\theta$ and smooths it with Gaussian noise on the input to yield the associated smoothed soft and hard classifiers

$$f^s(x) = \mathbb{E}_{\epsilon \sim N(0, \sigma^2 I_d)} f_\theta(x + \epsilon), \quad g(x) = \arg\max_{y \in \mathcal{Y}} f^s(x)_y,$$

where $f^s(x)_y$ denotes the $y$th component of the vector $f^s(x)$ and $\sigma$ is a hyperparameter. Cohen et al. (2019, Theorem 1) then gives, under certain conditions, a certified $\ell_2$-ball for a particular input $x \in \mathbb{R}^d$; namely, that $g(x + \delta) = g(x)$ for all $\|\delta\| < R$, where $R > 0$ is determined by the confidence of the smoothed classifier at $x$. We leverage this result for our approach and refer interested readers to Cohen et al. (2019) for additional details on the computation of the smoothing expectation and precise formula for $R$.

**Projected randomized smoothing.** Motivated by the relationships between robustness and dimensionality described in Section 1, we consider $p < d$ and let $P \colon \mathbb{R}^d \to \mathbb{R}^p$ be a projection into $\mathbb{R}^p$ defined by $P(x) = U^\intercal x$, where $U \in \mathbb{R}^{d \times p}$ is a semi-orthogonal matrix satisfying $U^\intercal U = I_p$. Similarly, we let the reconstruction $\tilde{P} \colon \mathbb{R}^p \to \mathbb{R}^d$ be defined by $\tilde{P}(\tilde{x}) = U\tilde{x}$. Throughout, we let $v_1, \ldots, v_{d-p} \in \mathbb{R}^d$ be an orthonormal basis for $\mathcal{N}(U^\intercal)$ and let $v_{d-p+1}, \ldots, v_d \in \mathbb{R}^d$ denote the orthonormal columns of $U$. In practice, we instantiate the columns of $U$ as the first $p$ principal components of a random subset of the training dataset, although our method and theory hold for any orthonormal set of vectors. With the dimension-reducing projection $P$ in place, we consider the classifier architecture consisting of the composition

$$f = f_\theta \circ \tilde{P} \circ P.$$

In particular, $f$ first uses $P$ to project inputs into the low-dimensional space $\mathbb{R}^p$ and then reconstructs the inputs in a lossy way using $\tilde{P}$ before feeding them through the classifier $f_\theta$. We generally finetune $f_\theta$ to account for the slight image corruption associated with the projection step.

---

[2] The zero-centered cube is used without loss of generality instead of $[0, 1]^d$ for notational convenience and compatibility with results from the mathematical literature.

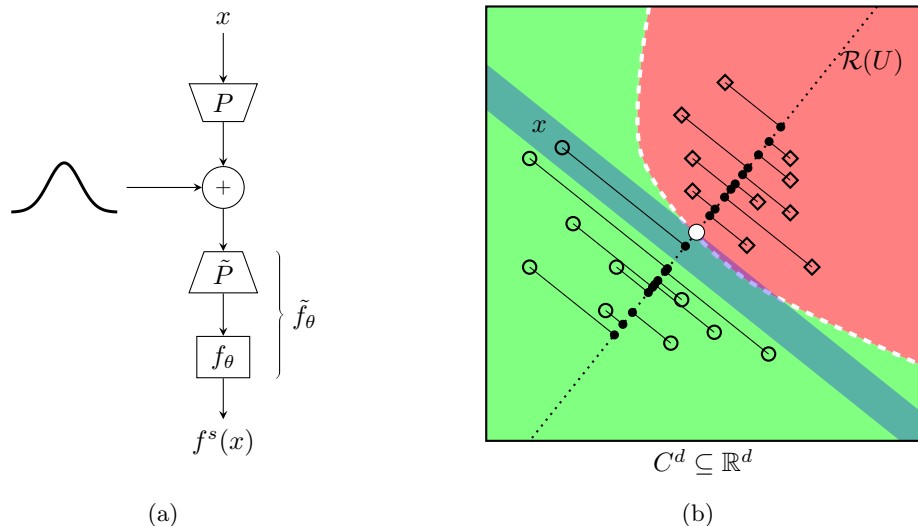

(a)                                                                    (b)

Figure 1: (a) Projected randomized smoothing architecture. Inputs $x$ are projected into low-dimensional space by $P$, smoothed with Gaussian noise, and then reconstructed by $\tilde{P}$ and classified by $f_\theta$. (b) Illustration of projected randomized smoothing for a binary classification task (circles vs. squares). The base classifier decision regions are shown in green and red. The white circle represents the smoothed decision boundary in $\mathbb{R}^p$, $p = 1$, with the projected subspace depicted by the dotted line and projected points depicted as solid dots. The blue area represents the certified region around $x$ in $\mathbb{R}^d$ of the projected randomized smoothing classifier $g$.

We now propose *projected randomized smoothing*, wherein randomized smoothing is performed in the compressed space $\mathbb{R}^p$. To do so, we define $\tilde{f}_\theta \colon \mathbb{R}^p \to [0,1]^c$ by $\tilde{f}_\theta = f_\theta \circ \tilde{P}$ so that $f = \tilde{f}_\theta \circ P$, and we smooth $\tilde{f}_\theta$ by adding Gaussian noise in its low-dimensional input space to obtain a new classifier $\tilde{f}_\theta^s \colon \mathbb{R}^p \to [0,1]^c$ defined by

$$\tilde{f}_\theta^s(\tilde{x}) = \mathop{\mathbb{E}}_{\epsilon \sim N(0, \sigma^2 I_p)} \tilde{f}_\theta(\tilde{x} + \epsilon). \tag{1}$$

The new overall smoothed soft classifier is then given by

$$f^s = \tilde{f}_\theta^s \circ P, \tag{2}$$

and its structure is illustrated in Figure 1a. The corresponding hard classifier is then given by the arg max of the soft classifier:[3]

$$g(x) = \operatorname*{arg\,max}_{y \in \mathcal{Y}} f^s(x)_y. \tag{3}$$

A graphical illustration of our approach for $d = 2$ is shown in Figure 1b. To summarize, classifying an input $x \in \mathbb{R}^d$ using projected randomized smoothing amounts to applying the mapping $x \mapsto g(x)$ defined by (1) through (3), and it is for $g$ that we seek to derive certified regions of the input space.

## 3   Robustness certificates

In this section, we construct certified regions for $g$ around arbitrary inputs $x$ in the high-dimensional space $\mathbb{R}^d$. The key idea is that $\tilde{f}_\theta^s$ is $\ell_2$-ball robust in the low-dimensional space $\mathbb{R}^p$, and the preimage of this ball in the original input space is then "large" as it includes the inputs in $\mathcal{N}(U^\intercal)$. We formalize the geometry of the certified region in Section 3.1 and introduce our metric of interest as the volume of the certified region restricted to the unit cube of feasible inputs. In Section 3.2, we provide a lower bound on this volume

---

[3]For ease of exposition, we assume throughout that all arg max yield singleton sets and therefore equality signs may be used unambiguously.

in high-dimensional spaces that involves solving an $\ell_\infty$-norm linear regression. Section 3.3 compares the asymptotic behavior of the volume of the certified region of $g$ with the standard $\ell_2$-ball certificates as the input dimension grows large. Finally, we discuss runtime and limitations in Section 3.4. For ease of exposition, all proofs are deferred to the appendices.

## 3.1 Characterizing the certified region geometry

In the following two propositions, we characterize the geometry of the projected randomized smoothing classifier $g$ in the high-dimensional input space $\mathbb{R}^d$ based on the certified $\ell_2$-robustness of the classifier $\tilde{f}_\theta^s$ in the low-dimensional projected space $\mathbb{R}^p$.

**Definition 1.** Let $\tilde{x} \in \mathbb{R}^p$ and $R \geq 0$. The classifier $\tilde{f}_\theta^s \colon \mathbb{R}^p \to [0,1]^c$ is said to be *certified at $\tilde{x}$ with radius R* if

$$\arg\max_{y \in \mathcal{Y}} \tilde{f}_\theta^s(\tilde{x} + \tilde{\delta})_y = \arg\max_{y \in \mathcal{Y}} \tilde{f}_\theta^s(\tilde{x})_y$$

for all $\tilde{\delta} \in \mathbb{R}^p$ satisfying $\|\tilde{\delta}\| \leq R$.

**Proposition 1.** *Let $x \in \mathbb{R}^d$ and $R \geq 0$. If $\tilde{f}_\theta^s$ is certified at $P(x) = U^\intercal x$ with radius $R$, then $g(x + \delta) = g(x)$ for all $\delta \in \Delta^U(R) \subseteq \mathbb{R}^d$, where*

$$\Delta^U(R) := \{\delta \in \mathbb{R}^d : \|U^\intercal \delta\| \leq R\}$$

**Proposition 2.** *Let $R \geq 0$. The certified region $\Delta^U(R)$ can be expressed as the Minkowski sum $\Delta^U(R) = B_p^U(R) + \mathcal{N}(U^\intercal)$, where $B_p^U(R) \subseteq \mathbb{R}^d$ is a p-dimensional ball embedded into $\mathcal{R}(U)$:*

$$B_p^U(R) := \{\beta_1 v_{d-p+1} + \cdots + \beta_p v_d : \|\beta\| \leq R, \ \beta \in \mathbb{R}^p\} .$$

Propositions 1 and 2 characterize the geometry of the certified region of our classifier $g$. Proposition 1 provides an easy-to-check condition for an input to lie in the certified region, while Proposition 2 formalizes the same geometry as a hypercylinder consisting of a low-dimensional sphere that is "extruded" along the nullspace of the projection $P$, allowing us to certify adversarial off-manifold inputs of potentially very large magnitude that are projected back onto the natural data manifold. Intuitively, the certified region $\Delta^U(R)$ is potentially much larger than an $\ell_2$-ball of radius $R$ in $\mathbb{R}^d$, as it captures perturbations in the nullspace of $U^\intercal$ whose dimensionality is large when $p \ll d$.

We note that the above characterization of the decision region geometry holds analogously for other norm ball certificates in the projected space (i.e., the $\ell_1$-ball certificates of Levine & Feizi (2021)). While the following theory is presented for the concrete case of $\ell_2$-ball certificates, it also applies to this more general setting. Concrete experiments with other certificates is an exciting line of future work.

## 3.2 Lower-bounding the certified region volume

To compare a standard $\ell_2$-ball certificate with our certified region $\Delta^U(R)$, which does not immediately come equipped with a notion of "radius," we adopt the perspective of recent works, e.g., Liu et al. (2019b); Eiras et al. (2021); Tecot (2021), by considering our metric of interest to be the volume of the certified region. One immediate issue is that the volume of $\Delta^U(R)$ is infinite since $\mathcal{N}(U^\intercal)$ is an unbounded subspace. To enable meaningful comparisons, we restrict ourselves to measuring the volume of $\Delta^U(R)$ contained in the cube $C^d = [-1/2, 1/2]^d$ of possible inputs. This amounts to computing the volume

$$V_d \left( C^d \cap \Delta_x^U(R) \right), \tag{4}$$

where we recall that $V_d$ measures $d$-dimensional volume in Euclidean space, and $\Delta_x^U(R) := \{x + \delta : \delta \in \Delta^U(R)\}$, with $R$ chosen such that $\tilde{f}_\theta^s$ is certified at $P(x)$ with radius $R$ so that $g(x') = g(x)$ for all $x' \in \Delta_x^U(R)$ by Proposition 1. Computing the volume in (4) is highly nontrivial, especially in high-dimensional input spaces. Instead, we develop a tractable lower bound on $V_d(C^d \cap \Delta_x^U(R))$ throughout the remainder of this section. Since $\Delta_x^U(R)$ contains affine subspaces, this derivation rests heavily on theory regarding cube-subspace

intersections in high dimensions. The most important result for our purposes comes from Vaaler (1979), which showed the following.

**Theorem 1.** *Let $S_k$ be a k-dimensional linear subspace of $\mathbb{R}^d$. Then $V_k(C^d \cap S_k) \geq 1$.*

This result proved Good's conjecture and generalized a previous result for the $k = d - 1$ case (Hensley, 1979). We begin with an extension of Theorem 1 to cubes of non-unit side length, and then to intersections with affine subspaces which do not necessarily contain the origin.

**Corollary 1.** *Let $S_k$ be a k-dimensional linear subspace of $\mathbb{R}^d$ and $rC^d$ be a zero-centered cube of side length $r > 0$. Then $V_k(rC^d \cap S_k) \geq r^k$.*

**Corollary 2.** *Let $x \in \mathbb{R}^d$ and let $S_k(x) \subseteq \mathbb{R}^d$ be the k-dimensional affine subspace*

$$S_k(x) = \left\{ x + \sum_{i=1}^k \alpha_i v_i : \alpha \in \mathbb{R}^k \right\}$$

*spanned by arbitrary vectors $v_1, \ldots, v_k$ and passing through $x$. Let $t \geq 0$ be the minimal $\ell_\infty$-norm of a point in $S_k(x)$:*

$$t := \inf_{x' \in S_k(x)} \|x'\|_\infty = \inf_{\alpha \in \mathbb{R}^k} \left\| x + \sum_{i=1}^k \alpha_i v_i \right\|_\infty. \tag{5}$$

*Then, for all $r > 2t$, it holds that $V_k(rC^d \cap S_k(x)) \geq (r - 2t)^k$.*

Corollary 2 generalizes Corollary 1 to affine subspaces. If $S_k(x)$ contains the origin, $t = 0$ and the bound from Corollary 1 is recovered. We are now ready to present the main result of this section.

**Theorem 2.** *Let $x \in C^d$, let $t$ be defined as in (5) with $k = d - p$, and let $R \in [0, 1/2 - t]$. If $\tilde{f}_\theta^s$ is certified at $P(x) = U^\intercal x$ with radius $R$, then*

$$V_d(C^d \cap \Delta_x^U(R)) \geq \frac{\pi^{p/2}}{\Gamma(\frac{p}{2} + 1)} R^p (1 - 2R - 2t)^{d-p}. \tag{6}$$

Notice that the lower bound given in Theorem 2 does not monotonically increase with the certified radius $R$ from the randomized smoothing performed in $\mathbb{R}^p$. Therefore, if the certified radius $R$ is large enough, we may be able to improve our lower bound on the volume $V_d(C^d \cap \Delta_x^U(R))$ by using a smaller certified radius (which is of course still valid), and in particular, we may choose the optimal such radius to use according to the following closed-form expression.

**Proposition 3.** *Let $t$ and $R$ be as in Theorem 2. The lower bound (6) is maximized as follows:*

$$r^* := \min \left\{ R, \frac{p(1 - 2t)}{2d} \right\} \in \arg\max_{r \in [0, R]} \frac{\pi^{p/2}}{\Gamma\left(\frac{p}{2} + 1\right)} r^p (1 - 2r - 2t)^{d-p}. \tag{7}$$

The overall certification procedure derived in this section is summarized in Algorithm 1. We note that our method inherits its ABSTAIN behavior from the original randomized smoothing Monte Carlo sampling scheme (Cohen et al., 2019); namely, we evaluate the certification confidence using many Gaussian-perturbed samples, and if the prediction or certification procedures do not resolve with a user-specified confidence, ABSTAIN is returned.

### 3.3 Asymptotic behavior of the volume bound

We briefly compare the volume lower bound (6) of the projected randomized smoothing certified region to that of a standard certified $\ell_2$-ball. The volume of a $d$-dimensional $\ell_2$-ball $B_d(R) := \{x \in \mathbb{R}^d : \|x\| \leq R\}$ of radius $R \geq 0$ is well-known (e.g., see Folland (1999, Theorem 2.44, Corollary 2.55)) to be

$$V_d(B_d(R)) = \frac{\pi^{d/2}}{\Gamma(\frac{d}{2} + 1)} R^d. \tag{8}$$

While the numerator of (8) scales exponentially in $d$, the denominator $\Gamma(\frac{d}{2} + 1)$ scales factorially, leading to tiny $\ell_2$-ball certified volumes in high-dimensional input spaces. By contrast, the denominator in our bound (6) scales factorially in the *projected dimension $p$*, where $p \ll d$. This suggests dramatic improvements in the volume of our certified regions: while the numerator in (6) might be *exponentially* smaller than that of (8), the denominator is smaller by a *factorial* factor. We thus expect the volumes of projected randomized smoothing to dominate at higher dimensions. We verify our analysis experimentally in Section 4.2 and illustrate some simulated certified volume ratios over a range of values for $p$ and $d$ in Appendix A.1.

---

**Algorithm 1** Prediction and certification

---

**def** PREDICT, CERTIFY as in Cohen et al. (2019)

**function** PROJECTPREDICT($f_\theta$, $U$, $\sigma$, $x$, $n$, $\alpha$)
  **def** $P(x) = U^\intercal x$, $\tilde{P}(\tilde{x}) = U\tilde{x}$
  **return** PREDICT($f_\theta \circ \tilde{P}, \sigma, P(x), n, \alpha$)

**function** PROJECTCERTIFY($f_\theta$, $U$, $\sigma$, $x$, $n_0$, $n$, $\alpha$)
  **def** $P(x) = U^\intercal x$, $\tilde{P}(\tilde{x}) = U\tilde{x}$, $(d, p) \leftarrow \text{shape}(U)$
  ABSTAIN, $\hat{c}_A, R \leftarrow$ CERTIFY($f_\theta \circ \tilde{P}, \sigma, P(x), n_0, n, \alpha$)
  **if** ABSTAIN **then return** ABSTAIN
  **compute** orthonormal basis $v_1, \ldots, v_{d-p}$ for $\mathcal{N}(U^\intercal)$
  **solve** the optimization

$$t \leftarrow \inf_{\alpha \in \mathbb{R}^{d-p}} \left\| x + \sum_{i=1}^{d-p} \alpha_i v_i \right\|_\infty \qquad \text{(Alg1)}$$

  **assign** $R \leftarrow \min\{R, p(1 - 2t)/(2d)\}$
  **compute** the certified volume lower bound

$$V \leftarrow \frac{\pi^{p/2}}{\Gamma(\frac{p}{2} + 1)} R^p (1 - 2R - 2t)^{d-p}$$

  **return** prediction $\hat{c}_A$ and volume bound $V$

---

## 3.4 Runtime and limitations

Our certification strategy has two additional computational steps outside of the PREDICT and CERTIFY subroutines from the conventional randomized smoothing method of Cohen et al. (2019). The first is a one-time computation of the principal components of the data that occurs at the beginning of training. The second is computing the $\ell_\infty$-regression in (Alg1), which we solve as a linear program using the standard epigraph formulation. For the CIFAR-10 and SVHN datasets considered in this work, the added runtime is comparable to the certification sampling step from Cohen et al. (2019). Namely, we found that the $\ell_\infty$-regression averaged around 16 seconds for CIFAR-10 and 19 seconds for SVHN.[4]

The number of variables and constraints in the optimization (Alg1) scales linearly with $d - p$. Since generally $p \ll d$, this makes the volume approximation of the certified region computationally intensive in high-dimensional spaces. We remark that it is still trivial to check whether any particular perturbation lies in the certified region using Proposition 1—it is just that computing a lower bound on the volume of this region for comparison purposes becomes more challenging. For a natural image dataset such as ImageNet, the analysis of Section 3.3 suggests that the certified region volume improvements would in fact be substantially larger than those for CIFAR-10. The main challenge to computationally verifying this conjecture lies in holding the optimization problem (Alg1) in memory, which is infeasible on our hardware for ImageNet-scale inputs.

---

[4]All experiments were run on a Ubuntu 20.04 virtual machine with 6 VCPUs, 56 GiB RAM, and a Tesla K80 GPU. Complete reproduction takes roughly 0.06 GPU years.

Further research in this vein would likely leverage techniques from the large-scale $\ell_\infty$-regression literature, e.g., Shen et al. (2014), and is outside the scope of this work.

## 4 Experiments

This section reports our experiments on the CIFAR-10 and SVHN datasets. We first demonstrate in Section 4.1 that networks are vulnerable to $\ell_\infty$-bounded attacks in the subspace of low-variance principal components, to which our architecture is provably robust. Section 4.2 then presents results comparing the volume of the projected randomized smoothing certified regions to a variety of baseline certified classifiers.

### 4.1 Vulnerability to low-variance PCA attacks

Consider perturbations $\delta \in \mathcal{N}(U^\intercal)$ contained in the span of a dataset's low-variance principal components, where we take $U$ to contain sufficient components to account for 99% of the dataset variance for CIFAR-10 and 95% for SVHN, which is more robust to low-variance subspace attacks due to its increased compressibility. Such a perturbation is known to be essentially orthogonal to the true data manifold, and therefore it is reasonable to expect a truly robust classifier to be invariant to small perturbations in $\mathcal{N}(U^\intercal)$. Our method is directly robust to such perturbations under the simple condition that we use fewer components in our initial projection step, as demonstrated in Proposition 2.

We now investigate whether this theoretical guarantee adds a degree of robustness over a typical neural network classifier. The answer is affirmative. Namely, we show that our subspace attack can attain a comparable attack success rate to a standard $\ell_\infty$-bounded projected gradient descent (PGD) attack, with roughly a four-fold increase in the size of the admissible $\ell_\infty$-ball.

Formally, consider a particular hard classifier $g$, to which we assume that our adversaries have white-box access, and take a specific input $x$ that $g$ classifies correctly. We first consider the standard projected gradient descent attack strategy $\mathrm{PGD}(x, \epsilon)$ which seeks to construct a perturbation $\|\delta\|_\infty \leq \epsilon$ such that $x + \delta \in C^d$ and $g(x + \delta) \neq g(x)$. As $x + \delta \in C^d$ if and only if $\|x + \delta\|_\infty \leq 1/2$, satisfying both $\ell_\infty$-norm constraints on $\delta$ is easily accomplished using clipping. Our routine $\textsc{SubspacePGD}(x, \epsilon)$ adds the additional constraint $\delta \in \mathcal{N}(U^\intercal)$. Note that finding a perturbation that satisfies $\delta \in \mathcal{N}(U^\intercal)$, $\|\delta\|_\infty \leq \epsilon$, and $x + \delta \in C^d$ is nontrivial, as projection onto one set generally removes an input from the other set. The precise details of our attack strategy are detailed in Appendix B.2.

For reference, we also consider $\textsc{RandMax}$ and $\textsc{RandUniform}$, which generate perturbations randomly on the boundary of and uniformly in the attack $\ell_\infty$-ball, respectively. We instantiate $g$ as the Wide ResNet considered in Yang et al. (2020) with the default hyperparameters and $\sigma = 0.15$ Gaussian noise augmentation during training. See Appendix B.5.1 for the attack hyperparameters.

Figure 2 demonstrates that unprotected classifiers are indeed vulnerable to adversarial perturbations in the subspace of low-variance principal components. Enlargements of the attack radius do not invalidate that these are true adversarial attacks, as the perturbed images in the third row of Figure 2b are still easily classified by a human. Furthermore, $\textsc{SubspacePGD}$ adversarial examples are substantially less perceptible than PGD attacks of the same magnitude, which tend to produce stronger visual distortions of the image, paralleling results from Shamir et al. (2021); take as a representative example the area around the frog's head in the second row of the third column in Figure 2b, compared with the same image perturbed by $\textsc{SubspacePGD}$ in the third row. The results for the SVHN dataset in Figure 2d are even more striking. This is likely because PGD attacks have access to high-variance principal components which convey the dataset information content. Despite visually appearing random, we establish in Figures 2a and 2c that the $\textsc{SubspacePGD}$ attack is significantly more successful than random-noise attacks of the same magnitude. These results suggest that undefended classifiers can be attacked in the subspace of low-variance principal components, to which projected randomized smoothing is provably robust by Proposition 2.

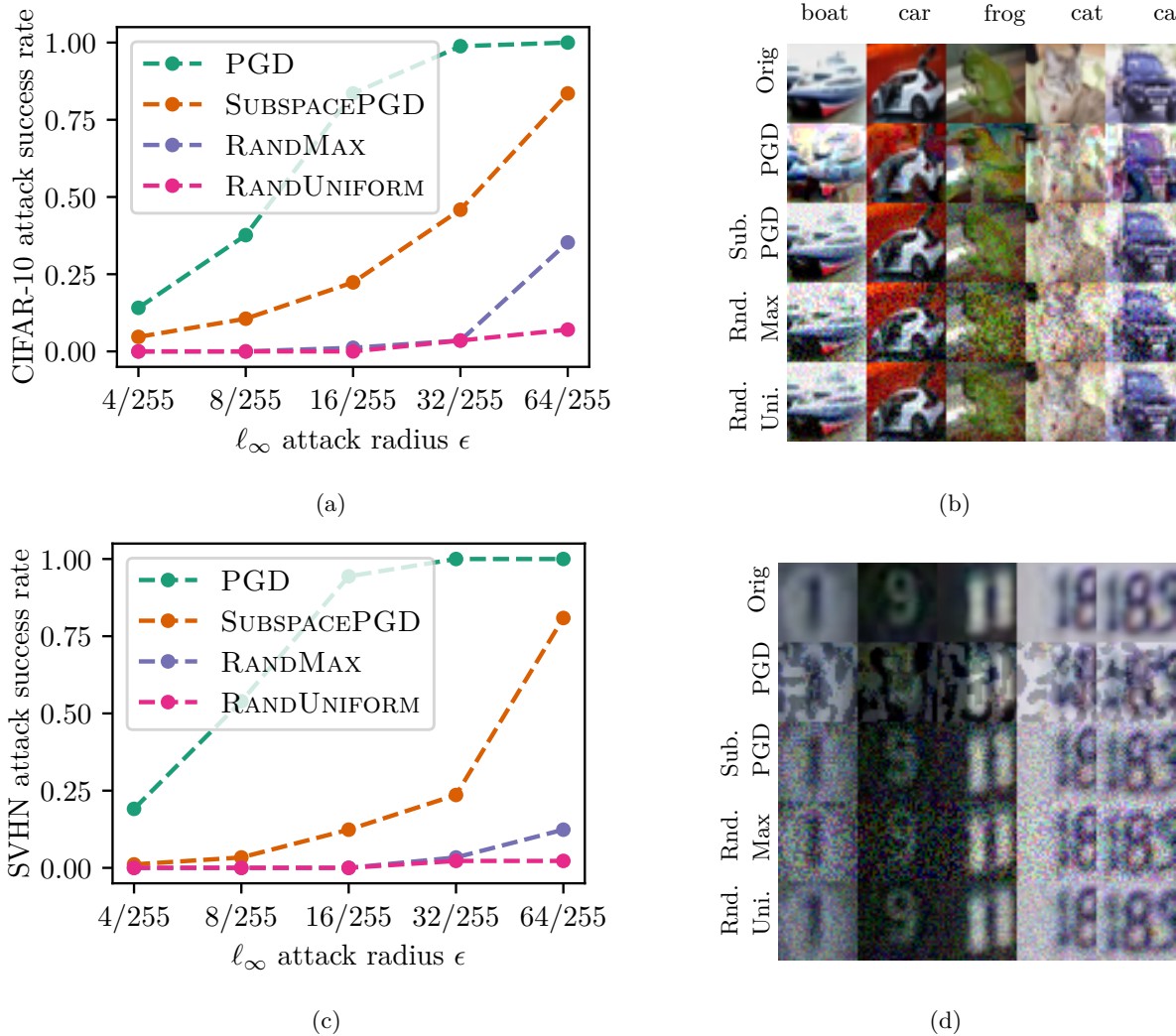

Figure 2: (a) CIFAR-10 adversarial attack success rates for the PGD, SUBSPACEPGD, and random attack strategies. (b) Perturbation examples for CIFAR-10 with an attack radius of $\epsilon = 32/255$. The top row represents the original image. (c) SVHN aversarial attack success rates for the PGD, SUBSPACEPGD, and random attack strategies. (d) Perturbation examples for SVHN with an attack radius of $\epsilon = 32/255$.

## 4.2 Certified region comparison

Having established that the certified region of projected randomized smoothing provides a meaningful robustness improvement against low-variance principal component attacks, we now compare the volume of our certified region with several baselines. Namely, we evaluate the $\ell_2$-balls of Cohen et al. (2019) (denoted RS), the $\ell_1$- and $\ell_\infty$-balls of Yang et al. (2020) (denoted RS4A $- \ell_1$ and RS4A $- \ell_\infty$, respectively), and the anisotropic ellipsoids of Eiras et al. (2021) (denoted ANCER), without use of the associated memory module.

Some additional remarks on the inclusion of Eiras et al. (2021) are warranted. As noted in Súkeník et al. (2021), without the inclusion of the memory module, the local certificate optimization technique in Eiras et al. (2021) yields overly optimistic and mathematically incorrect certificates as the smoothing distribution varies between inputs. The work Eiras et al. (2021) corrects this with the use of a memory module that records previous inputs to ensure compatibility of the smoothing certificates. However, this results in a classifier that is dependent on the input order and adds ambiguity about what classifier is actually being certified, as the smoothed classifier is modified at test time after each input. We therefore discard the memory module and

report the certified volume at each point as if the locally optimized smoothing distribution were being used globally. This yields an upper bound on the certified volume of any data-dependent anisotropic ellipsoidal smoothing method and is thus a very strong baseline to compare against.

Our results are summarized in Figure 3 and Table 1. We achieve state-of-the-art median certified volumes, easily outperforming standard randomized smoothing and even the optimistic ANCER baseline by 706 and 2453 *orders of magnitude* on CIFAR-10 and SVHN, respectively. The larger improvement on SVHN is attributable to the higher compressibility of the dataset. Figure 6 in Appendix B.3 suggests that our performance derives from the added robustness of our method against low-variance features, as the radii of the projected-space certified balls are similar to those of standard randomized smoothing in the input space. This further validates the asymptotic dimension analysis in Section 3.3. Note that although the ANCER baseline achieves higher accuracy at smaller volumes, its certificates are mathematically invalid (Súkeník et al., 2021), and our method significantly outperforms ANCER at larger volumes.

Figure 3b examines the CIFAR-10 certified accuracy curves over a range of choices for the dimensionality $p$ of the compressed space. For large $p$, image reconstruction is near-perfect as $p = 620$ covers $99\%$ of variance in the CIFAR-10 dataset. Thus, methods with $p \geq 300$ have comparable accuracy at small regions, with the certified volumes increasing as the dimensionality of the projected space decreases, corroborating the discussion in Section 3.3. We are therefore able to increase the robustness of our classifier to disturbances that are normal to the manifold with only a $2\%$ drop in accuracy (Table 1a). Figure 3d presents similar results for the SVHN dataset. Note that the due to the compressibility of the dataset, fewer principal components are required to achieve high accuracy.

The hyperparameter $p$ introduces a mild tradeoff between clean accuracy and certified volume; if $p$ is chosen to be very small, the projected images may be too corrupted to classify, while if $p$ is chosen to be very large, certified volume may suffer. However, as Figures 3b and 3d suggest, our method's certified volumes comfortably outperform those of standard randomized smoothing for a large range of $p$, indicating that this choice is not particularly sensitive. A practical heuristic for choosing $p$ involves making $p$ just large enough to reconstruct images with high fidelity—roughly corresponding to PCA components that explain $95\%$ to $99\%$ of the dataset variance. If desired, a small, localized sweep of $p$ around this initial choice can be used to further optimize the hyperparameter depending on the experimentalist's target metrics (e.g., clean accuracy, median certified volume, other metrics, or some combination). In any case, we emphasize that the parameter choice is quite robust and any additional tuning is likely to result in minimal gains as compared to the practical heuristic. We select $p = 450$ for the CIFAR-10 experiment in Figure 3a and $p = 150$ for SVHN.

Table 1: Quantitative representation of the data in Figure 3. The first column reports the smoothed classifier clean accuracy for each method and the second column reports the median certified volume for correctly classified samples. We use the median instead of the mean due to the log-scaled nature of our data.

(a) CIFAR certification performance.

| | Accuracy | Median cert. vol. ($\log_{10}$) |
|---|---|---|
| PROJECTEDRS | $85.8\%$ | $\mathbf{-3175}$ |
| RS | $\mathbf{87.8}\%$ | $-4377$ |
| ANCER | $87.4\%$ | $-3881$ |
| RS4A $- \ell_1$ | $83.8\%$ | $-9573$ |
| RS4A $- \ell_\infty$ | $85.4\%$ | $-6102$ |

(b) SVHN certification performance.

| | Accuracy | Median cert. vol. ($\log_{10}$) |
|---|---|---|
| PROJECTEDRS | $91.4\%$ | $\mathbf{-1578}$ |
| RS | $92.6\%$ | $-4280$ |
| ANCER | $91.2\%$ | $-4031$ |
| RS4A $- \ell_1$ | $\mathbf{93.0}\%$ | $-9573$ |
| RS4A $- \ell_\infty$ | $92.6\%$ | $-6171$ |

## 5 Conclusion

Motivated by the manifold hypothesis, we consider a classifier architecture that first projects onto a principal component approximation of the data manifold and then applies randomized smoothing in the low-dimensional projected space. This yields a precise characterization of the input-space certified region as capturing

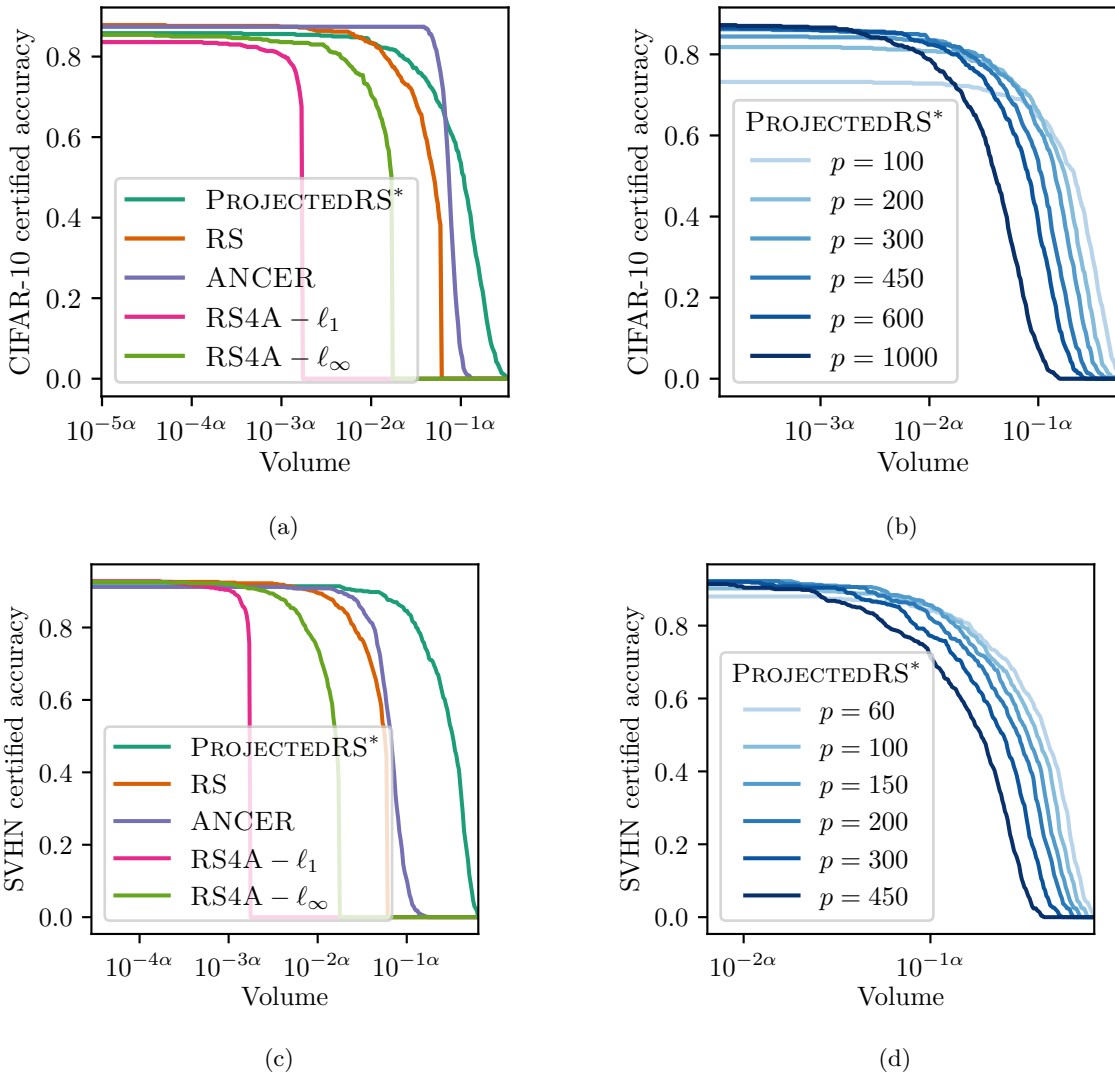

Figure 3: (a) Certified region volumes for CIFAR-10, with our method highlighted by an asterisk. Here $\alpha \approx 3465$ is a scaling constant corresponding to the $d$-dimensional unit ball volume; i.e. $V_d(B_d(1)) = 10^{-\alpha}$. (b) CIFAR-10 certified region volumes while varying the projected space dimension $p$ for our method. (c) Certified region volumes for SVHN. (d) SVHN certified region volumes while varying $p$.

disturbances in the projection nullspace. We interpret this as a certifiable robustification against vulnerable features that are irrelevant to the dataset information content as they are normal to the data manifold. We show that unprotected classifiers, unlike our method, are vulnerable to such perturbations by explicitly constructing adversarial examples in the span of the low-variance principal components. We prove a volumetric lower bound on the intersection of our certified region with the unit cube of feasible inputs and derive two additional ways to tighten the bound: one which involves solving an $\ell_\infty$-regression problem and another which is a closed-form radius adjustment.

Comparing against state-of-the-art $\ell_1$-, $\ell_2$-, $\ell_\infty$-, and anisotropic baselines shows that our classifier produces certified regions with many orders of magnitude greater volume. This confirms an asymptotic analysis that shows that our method's certified volumes decay factorially in the low dimension of the *projected space*, while competing methods decay factorially in the high dimension of the *input space*. Future research directions include examining more sophisticated dimensionality reduction techniques while maintaining certified guarantees for projected points in the original input space.

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

## A    Supplementary material for Section 3 (Robustness certificates)

**Proposition 1.** *Let $x \in \mathbb{R}^d$ and $R \geq 0$. If $\tilde{f}_\theta^s$ is certified at $P(x) = U^\mathsf{T} x$ with radius $R$, then $g(x + \delta) = g(x)$ for all $\delta \in \Delta^U(R) \subseteq \mathbb{R}^d$, where*

$$\Delta^U(R) := \{\delta \in \mathbb{R}^d : \|U^\mathsf{T} \delta\| \leq R\}$$

*Proof.* Let $\delta \in \Delta^U(R)$. Then

$$g(x + \delta) = \arg\max_{y \in \mathcal{Y}} \tilde{f}_\theta^s(P(x + \delta))_y = \arg\max_{y \in \mathcal{Y}} \tilde{f}_\theta^s(P(x) + U^\mathsf{T} \delta)_y.$$

Since $\|U^\mathsf{T}\delta\| \leq R$ by definition of $\Delta^U(R)$ and $\tilde{f}_\theta^s$ is certified at $P(x)$ with radius $R$, we have that

$$g(x + \delta) = \arg\max_{y \in \mathcal{Y}} \tilde{f}_\theta^s(P(x))_y = g(x).$$

$\square$

**Proposition 2.** *Let $R \geq 0$. The certified region $\Delta^U(R)$ can be expressed as the Minkowski sum $\Delta^U(R) = B_p^U(R) + \mathcal{N}(U^\mathsf{T})$, where $B_p^U(R) \subseteq \mathbb{R}^d$ is a p-dimensional ball embedded into $\mathcal{R}(U)$:*

$$B_p^U(R) := \{\beta_1 v_{d-p+1} + \cdots + \beta_p v_d : \|\beta\| \leq R, \ \beta \in \mathbb{R}^p\}.$$

*Proof.* Let $y = y_1 + y_2$ with $y_1 \in B_p^U(R)$ and $y_2 \in \mathcal{N}(U^\mathsf{T})$. Then

$$\|U^\mathsf{T} y\| = \|U^\mathsf{T} y_1\| = \|\beta\| \leq R,$$

so $y \in \Delta^U(R)$.

On the other hand, let $y \in \Delta^U(R)$ as defined in Proposition 1. We can decompose $y = y_1 + y_2$ for $y_1 \in \mathcal{R}(U)$ and $y_2 \in \mathcal{N}(U^\mathsf{T})$. Then there exists $\beta \in \mathbb{R}^p$ such that $y_1 = U\beta = \sum_{i=d-p+1}^n \beta_{i-d+p} v_i$, so $\|U^\mathsf{T} y_1\| = \|\beta\|$ and therefore $\|\beta\| \leq R$. $\square$

**Corollary 1.** *Let $S_k$ be a k-dimensional linear subspace of $\mathbb{R}^d$ and $rC^d$ be a zero-centered cube of side length $r > 0$. Then $V_k(rC^d \cap S_k) \geq r^k$.*

*Proof.* Note that

$$\begin{aligned}
rC^d \cap S_k &= \{x \in \mathbb{R}^d : \|x\|_\infty \leq r/2, \ x \in S_k\} \\
&= \{rx \in \mathbb{R}^d : \|rx\|_\infty \leq r/2, \ rx \in S_k\} \\
&= \{rx \in \mathbb{R}^d : \|x\|_\infty \leq 1/2, \ x \in S_k\},
\end{aligned}$$

since $x \in S_k$ if and only if $rx \in S_k$, by linearity of $S_k$. This is now equivalent to the set $r(C^d \cap S_k)$, and we have scaled our $k$-dimensional subset by a uniform factor $r$. Therefore, $V_k(rC^d \cap S_k) = V_k(r(C^d \cap S_k)) = r^k V_k(C^d \cap S_k)$ by Folland (1999, Theorem 2.44). Thus, by Theorem 1, we have $V_k(rC^d \cap S_k) \geq r^k$. $\square$

**Corollary 2.** *Let $x \in \mathbb{R}^d$ and let $S_k(x) \subseteq \mathbb{R}^d$ be the k-dimensional affine subspace*

$$S_k(x) = \left\{x + \sum_{i=1}^k \alpha_i v_i : \alpha \in \mathbb{R}^k\right\}$$

*spanned by arbitrary vectors $v_1, \ldots, v_k$ and passing through $x$. Let $t \geq 0$ be the minimal $\ell_\infty$-norm of a point in $S_k(x)$:*

$$t := \inf_{x' \in S_k(x)} \|x'\|_\infty = \inf_{\alpha \in \mathbb{R}^k} \left\|x + \sum_{i=1}^k \alpha_i v_i\right\|_\infty. \tag{5}$$

*Then, for all $r > 2t$, it holds that $V_k(rC^d \cap S_k(x)) \geq (r - 2t)^k$.*

*Proof.* First, notice that the infimum in (5) is attained since $\|\cdot\|_\infty$ is continuous and coercive, and $S_k(x)$ is closed in the standard topology on $\mathbb{R}^d$ (Bertsekas, 2016). Let $x^* \in S_k(x)$ be a point that attains the infimum in (5) so that $\|x^*\|_\infty = t$. If $r > 2t$, then $x^*$ is contained in the interior of $rC^d$. In this case, we can construct a nonempty cube centered at $x^*$ with side lengths $r - 2t > 0$ that is contained in $rC^d$. Now, the plane $S_k(x)$ passes through $x^*$, and therefore Corollary 1 yields the result since volume is preserved under translation (Folland, 1999, Theorem 2.42). $\qquad\square$

**Theorem 2.** *Let $x \in C^d$, let $t$ be defined as in (5) with $k = d - p$, and let $R \in [0, 1/2 - t]$. If $\tilde{f}_\theta^s$ is certified at $P(x) = U^\mathsf{T} x$ with radius $R$, then*

$$V_d(C^d \cap \Delta_x^U(R)) \geq \frac{\pi^{p/2}}{\Gamma(\frac{p}{2} + 1)} R^p (1 - 2R - 2t)^{d-p}. \tag{6}$$

*Proof.* The characterization of $\Delta^U(R)$ in Proposition 2 yields

$$\Delta_x^U(R) = B_p^U(R) + S_{d-p}^{\mathcal{N}(U^\mathsf{T})}(x),$$

where

$$S_{d-p}^{\mathcal{N}(U^\mathsf{T})}(x) := \{x\} + \mathcal{N}(U^\mathsf{T})$$

is the affine subspace of $\mathbb{R}^d$ spanned by $\mathcal{N}(U^\mathsf{T})$ and passing through $x$, which has dimension $d - p$. Therefore, the following is an inner-approximation of $\Delta_x^U(R)$:

$$\tilde{\Delta}_x^U(R) := B_p^U(R) + \left( (1 - 2R)C^d \cap S_{d-p}^{\mathcal{N}(U^\mathsf{T})}(x) \right) \subseteq B_p^U(R) + S_{d-p}^{\mathcal{N}(U^\mathsf{T})}(x) = \Delta_x^U(R).$$

If we can show that $\tilde{\Delta}_x^U(R) \subseteq C^d$, then $\tilde{\Delta}_x^U(R) \subseteq C^d \cap \Delta_x^U(R)$, in which case the volume of $\tilde{\Delta}_x^U(R)$ will lower-bound the volume of $C^d \cap \Delta_x^U(R)$. To prove that this holds, let $y = y_1 + y_2 \in \tilde{\Delta}_x^U(R)$ with $y_1 \in B_p^U(R)$ and $y_2 \in (1 - 2R)C^d \cap S_{d-p}^{\mathcal{N}(U^\mathsf{T})}(x)$. Then

$$\|y\|_\infty \leq \|y_1\|_\infty + \|y_2\|_\infty \leq R + \frac{1 - 2R}{2} = \frac{1}{2},$$

by the fact that $\|y_1\|_\infty \leq \|y_1\| = \|U\beta\| = \|\beta\|$ for some $\beta \in \mathbb{R}^p$ with $\|\beta\| \leq R$ due to the semi-orthogonality of $U$, and by the fact that $y_2 \in (1 - 2R)C^d$. Therefore, indeed it holds that $\tilde{\Delta}_x^U(R) \subseteq C^d$. Thus, all that remains is to lower-bound $V_d(\tilde{\Delta}_x^U(R))$. To this end, notice that $B_p^U(R) \subseteq \mathcal{R}(U)$ and $(1 - 2R)C^d \cap S_{d-p}^{\mathcal{N}(U^\mathsf{T})}(x) \subseteq \{x\} + \mathcal{N}(U^\mathsf{T})$, so $B_p^U(R)$ and $(1 - 2R)C^d \cap S_{d-p}^{\mathcal{N}(U^\mathsf{T})}(x)$ are contained in orthogonal affine subspaces, and therefore $V_d(\tilde{\Delta}_x^U(R)) = V_p(B_p^U(R)) V_{d-p}((1 - 2R)C^d \cap S_{d-p}^{\mathcal{N}(U^\mathsf{T})}(x))$. The $p$-dimensional volume of the embedded ball $\ell_2$-ball $B_p^U(R)$ is well-known (e.g., see Folland (1999, Theorem 2.44, Corollary 2.55)) to be

$$V_p(B_p^U(R)) = \frac{\pi^{p/2}}{\Gamma(\frac{p}{2} + 1)} R^p.$$

On the other hand, since $2R < 1 - 2t$, it holds that $1 - 2R > 2t$. Hence Corollary 2 gives that the $(d - p)$-dimensional volume of $(1 - 2R)C^d \cap S_{d-p}^{\mathcal{N}(U^\mathsf{T})}(x)$ is lower-bounded as

$$V_{d-p}((1 - 2R)C^d \cap S_{d-p}^{\mathcal{N}(U^\mathsf{T})}(x)) \geq (1 - 2R - 2t)^{d-p}.$$

Therefore,

$$V_d(\tilde{\Delta}_x^U(R)) \geq \frac{\pi^{p/2}}{\Gamma(\frac{p}{2} + 1)} R^p (1 - 2R - 2t)^{d-p},$$

which concludes the proof. $\qquad\square$

**Proposition 3.** *Let $t$ and $R$ be as in Theorem 2. The lower bound (6) is maximized as follows:*

$$r^* := \min\left\{ R, \frac{p(1 - 2t)}{2d} \right\} \in \underset{r \in [0, R]}{\arg\max} \frac{\pi^{p/2}}{\Gamma\left(\frac{p}{2} + 1\right)} r^p (1 - 2r - 2t)^{d-p}. \tag{7}$$

*Proof.* It suffices to maximize $h(r) := r^p (1 - 2r - 2t)^{d-p}$ over $r \in [0, R]$. The gradient of $h$ vanishes at points satisfying

$$
\begin{aligned}
\frac{dh}{dr}(r) &= pr^{p-1} (1 - 2r - 2t)^{d-p} - 2(d-p)r^p (1 - 2r - 2t)^{d-p-1} \\
&= r^{p-1} (1 - 2r - 2t)^{d-p-1} \left( p (1 - 2r - 2t) - 2(d-p)r \right) \\
&= r^{p-1} (1 - 2r - 2t)^{d-p-1} \left( p - 2pt - 2dr \right) \\
&= 0.
\end{aligned}
$$

The set of all critical points satisfying this polynomial equation is $\left\{ 0, \frac{p(1-2t)}{2d}, 1/2 - t \right\}$. Notice that $0 < \frac{p(1-2t)}{2d} < \frac{p(1-2t)}{2p} = 1/2 - t$, and that $\frac{dh}{dr}(r) \geq 0$ for all $r \in \left[ 0, \frac{p(1-2t)}{2d} \right]$ whereas $\frac{dh}{dr}(r) \leq 0$ for all $r \in \left[ \frac{p(1-2t)}{2d}, 1/2 - t \right]$. Hence, $h$ is unimodal on $[0, 1/2 - t]$ with the maximizer $\frac{p(1-2t)}{2d}$. Therefore, if $R < \frac{p(1-2t)}{2d}$, then $h$ is monotone increasing on the feasible interval $[0, R]$, which implies that the right endpoint $r^* = R$ is a maximizer of (7). On the other hand, if $R \geq \frac{p(1-2t)}{2d}$, then $\frac{p(1-2t)}{2d}$ is contained in the feasible interval $[0, R]$, and thus $r^* = \frac{p(1-2t)}{2d}$ is a maximizer of (7). $\qquad \square$

As an aside, we note that the certified region of our method contains an $\ell_2$-ball of radius usually comparable to that of standard randomized smoothing, although in general the certified region of Propositions 1 and 2 will be much larger as it captures the null space of the projection operator. We nevertheless include the following simple result for completeness.

**Proposition 4.** *Let $x \in \mathbb{R}^d$ and $R \geq 0$. If $\tilde{f}_\theta^s$ is certified at $P(x) = U^\intercal x$ with radius $R$, then $g(x + \delta) = g(x)$ for all $\delta \in B_d(R) := \{ x \in \mathbb{R}^d : \|x\| \leq R \}$.*

*Proof.* Let $\delta \in B_d(R)$. Then, since $U^\intercal$ is a semi-orthogonal matrix, its $\ell_2$-induced operator norm $\|U^\intercal\|$ is less than or equal to 1. Thus, $\|U^\intercal \delta\| \leq \|U^\intercal\| \|\delta\| \leq R$. Therefore, $B_d(R) \subseteq \Delta^U(R)$. $\qquad \square$

## A.1 Finite-dimensional volume analysis

We complement our asymptotic certified volume comparison in Section 3.3 with a simple finite-dimensional sweep over input dimension $d$ and projected dimension $p$, the results of which are shown in Figure 4.

Here, we fix $R = 0.5$ and $t = 0.4$ as typical values for natural image datasets and sweep over a range of choices of $p$ and $d$. We assume that the low-dimensional and high-dimensional certified radii are similar, as indicated by Figure 6. The plotted values provide the ratio of our certified volume from Theorem 2 to the volume of a standard $\ell_2$-ball, as given in Section 3.3, e.g., a value of $\times 30000$ indicates that the volume of our certified region is 30000 times greater. This ratio grows rapidly as $d$ increases due to the factorial growth noted in Section 3.3.

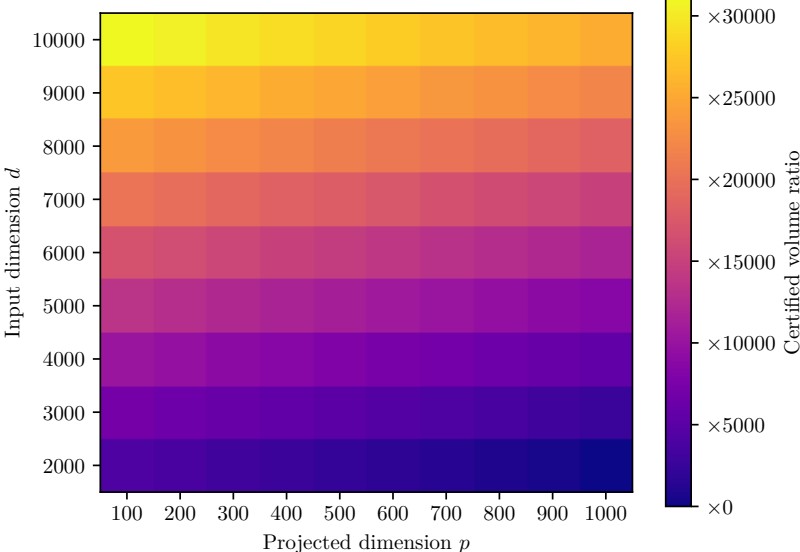

Figure 4: Ratio of projected randomized smoothing certified volume versus standard randomized smoothing certified volume for simulated values.

# B Supplementary material for Section 4 (Experiments)

## B.1 Random projections ablation

We compare the PCA projections used in our experiments with projection onto a random subspace for CIFAR-10. As reconstruction fidelity is much poorer for random subspaces, we expect to need many more components to achieve a comparable clean accuracy to PCA projections. We correspondingly adjust the number of components to $p = 1500$ in Figure 5 for random projections, and retain $p = 450$ PCA components. As with the hyperparameter sweeps, we use $n = 10^4$ smoothing samples. Figure 5 shows that projecting onto the PCA basis generally provides superior certificates. We also plot standard randomized smoothing for reference.

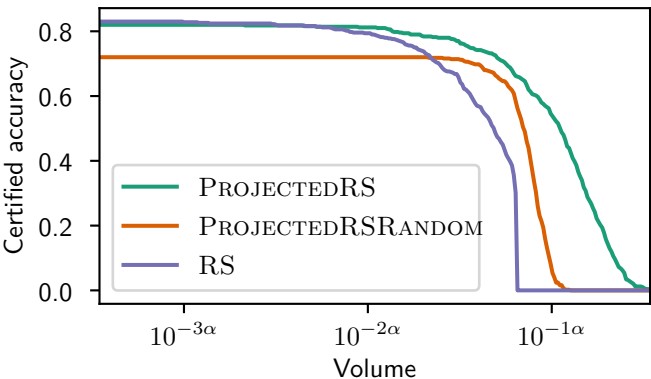

Figure 5: Ablation test comparing PCA basis projections with projection onto a random orthonormal set of vectors.

## B.2 Subspace attack procedure

A typical PGD attack constructs adversarial examples by iteratively perturbing the image along the gradient of the loss and projecting onto the unit cube of feasible inputs:

$$x^{(i+1)} = P_{C^d}\left(x + P_{C_\epsilon^d}\left(\alpha\,\mathrm{sign}\left(\nabla_\delta \mathcal{L}(x^{(i)} + \delta, y) - x^{(i)}\right)\right)\right),$$

where $x^{(i)}$ is the $i$th iterate of the PGD attack, $\mathcal{L}$ is the loss function, $\mathrm{sign}(\cdot)$ is the element-wise sign operator, $\alpha$ is the step size hyperparameter, $P_{C^d}$ projects a point in $\mathbb{R}^d$ onto $C^d$ by simple clipping, and $P_{C_\epsilon^d}$ is defined similarly for the zero-centered cube of sidelength $2\epsilon$. We initialize $x^{(1)} = x$, where $(x, y)$ are the original input and label from the dataset.

We desire a final perturbation $\delta$ such that $x + \delta \in C^d$, $\delta \in C_\epsilon^d$, and $\delta \in \mathcal{N}(U^\intercal)$. We first parameterize our perturbation in terms of the vectors $v_1, \ldots, v_{d-p}$ spanning $\mathcal{N}(U^\intercal)$. Stacking these vectors columnwise to yield $V \in \mathbb{R}^{d \times d-p}$, we can express our pertubation as $\delta = V\delta_V$ with $\delta_V \in \mathbb{R}^{d-p}$. We then iterate over our parameterized perturbations $\delta_V^{(i)}$, first solving for our "target" perturbation

$$\left(\delta_V^{(i+1)}\right)^* = \delta_V^{(i)} + \alpha\,\mathrm{sign}\left(\nabla_{\delta'}\mathcal{L}(x + V(\delta_V^{(i)} + \delta'), y)\right).$$

We then project the perturbation to satisfy the $\ell_\infty$-constraints, which takes the form of a quadratic program:

$$\begin{aligned}
\underset{\delta_V \in \mathbb{R}^{d-p}}{\text{minimize}} \quad & \left\|V\left(\delta_V^{(i+1)}\right)^* + V\delta_V\right\|_2^2 \\
\text{subject to} \quad & \|V\delta_V\|_\infty \leq \epsilon, \\
& \|x + V\delta_V\|_\infty \leq 1/2.
\end{aligned}$$

This program is always feasible with $\delta_V = 0$, and its solution satisfies our requirements for each iteration of the attack procedure.

## B.3 CIFAR-10 additional results

Here we present an additional plot comparing the radii of the *low-dimensional* projected randomized smoothing balls to the radii of standard randomized smoothing balls in the high-dimensional space. These are very similar, suggesting that the increase in the volume of the certified region comes from the "extrusion" of the ball, which amounts to added robustness against unnecessary features that are removed in the initial projection step. For the anisotropic ANCER method, we report the geometric mean of the radii along each coordinate axis, which Eiras et al. (2021) defines to be the "proxy radius." We only compare methods with $\ell_2$-based certified regions in this plot.

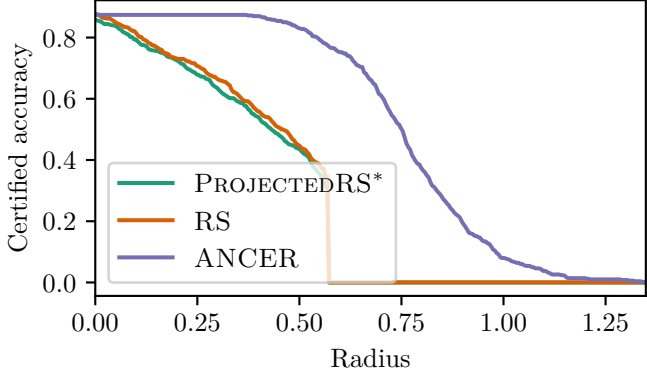

Figure 6: Certified radii on the CIFAR-10 dataset.

### B.4 Empirical manifold-perpendicular robustness

While the focus of our work is certified robustness, for expository purposes we briefly compare the empirical performance of our model against that of standard randomized smoothing on an approximately manifold-perpendicular threat model. Specifically, following a projected gradient descent iteration, we clamp the high-variance components of the perturbation to have norm at most $R$, where $R$ is a parameter that we sweep over in the table below. An $R$ of zero corresponds to a manifold-perpendicular attack, and as $R$ increases perturbations are allowed a larger on-manifold component. We also range over the perturbation magnitude $\epsilon$. The reported numbers are CIFAR-10 empirical accuracies in percentages, with our method highlighted in bold.

Table 2: CIFAR-10 empirical robust accuracy percentages subject to an approximately manifold-perpendicular threat model, with randomized smoothing in non-bold font and our method in bold.

| $\epsilon$ \ $R$ | 0.0 | 0.25 | 0.5 | 1.0 |
|---|---|---|---|---|
| $8/255$ | $76 - \mathbf{79}$ | $65 - \mathbf{69}$ | $54 - \mathbf{58}$ | $34 - \mathbf{35}$ |
| $16/255$ | $71 - \mathbf{79}$ | $60 - \mathbf{69}$ | $50 - \mathbf{58}$ | $28 - \mathbf{35}$ |
| $32/255$ | $59 - \mathbf{79}$ | $50 - \mathbf{69}$ | $39 - \mathbf{58}$ | $20 - \mathbf{35}$ |

We evaluate over 500 test images and execute smoothing for 100 samples at each input; while this is fewer than what is typically used for certification, it is a standard number of samples for the prediction problem (Cohen et al., 2019).

As seen in Table 2, our method consistently attains higher empirical robust accuracy than conventional randomized smoothing across all $\epsilon$-$R$ pairs tested. The advantages of our projection-based method are most apparent for small $R$, where the underlying geometry of the data distribution—which conventional randomized smoothing is naive to—is most influential, as well as for large $\epsilon$, where an attacker is able to create large off-manifold perturbations of the input data. As our theory suggests, we find that the empirical robust accuracy of our model remains constant over increasing $\epsilon$ for a fixed value of $R$, indicating that our model is not sensitive to increases in the off-manifold components of an attack. On the contrary, we see that, for fixed $R$, as $\epsilon$ increases, the empirical robust accuracy of conventional randomized smoothing decreases, meaning that conventional randomized smoothing is empirically sensitive to moving further away from the natural data manifold, an attack strategy that we have now shown our method to be both theoretically and empirically robust to.

### B.5 Hyperparameter selection

To maintain consistency, all networks we consider are Wide ResNets pretrained with various noise distributions using the code provided by Yang et al. (2020). For networks composed with an initial projection, we finetune the network with a learning rate or 0.001, momentum of 0.9, and weight decay of 0.0005 for 20 epochs, decaying the learning rate by a multiplicative factor of 0.95 per epoch.

#### B.5.1 Attack hyperparameters

We kept the attack hyperparameters fixed across both CIFAR-10 and SVHN. For the PGD attack, we use the torchattacks library with 40 steps and step size $\alpha = 2/255$ Kim (2020). We lowered this to 5 steps with $\alpha = \epsilon/4$ for SUBSPACEPGD due to the solve time of the projection step.

#### B.5.2 CIFAR-10 certification hyperparameters

We include the results of our hyperparameter sweeps for CIFAR-10 in Figure 7. For the RS4A $- \ell_1$ method, we used uniform noise and stability training to reproduce the state-of-the-art result from Yang et al. (2020). The RS4A $- \ell_\infty$ sweep used Guassian noise, which we found to perform better in practice. Our sweep over

the ANCER learning rate held the number of steps and regularization weight fixed at their defaults of 900 and 2, respectively. All sweeps were performed over 500 random test samples besides ANCER which was run over 100 samples due to the method's high computational burden.

The results from these sweeps informed the choice of hyperparameters in Figure 3a. Namely, we choose $\sigma = 0.25$ for our RS4A $- \ell_1$ baseline and $\sigma = 0.15$ for our RS4A $- \ell_\infty$ baseline, as the clean accuracy drops substantially for higher variances without approaching comparable certified volume to the other methods considered. We choose a learning rate of 0.01 for ANCER and $p = 450$ components for projected randomized smoothing. All experiments in the hyperparameter sweeps were performed with the smoothing hyperparameters of $n_0 = 100$ samples to guess the smoothed class, $n = 10^4$ samples to lower-bound the smoothed class probability, and a confidence of $\alpha = 0.001$. For reproducing the final results in Figure 3a we increased $n$ to $10^6$ as is standard (Cohen et al., 2019) and used 500 test samples to generate the plots. The attack experiment illustrated in Figure 2a was conducted over 100 test samples.

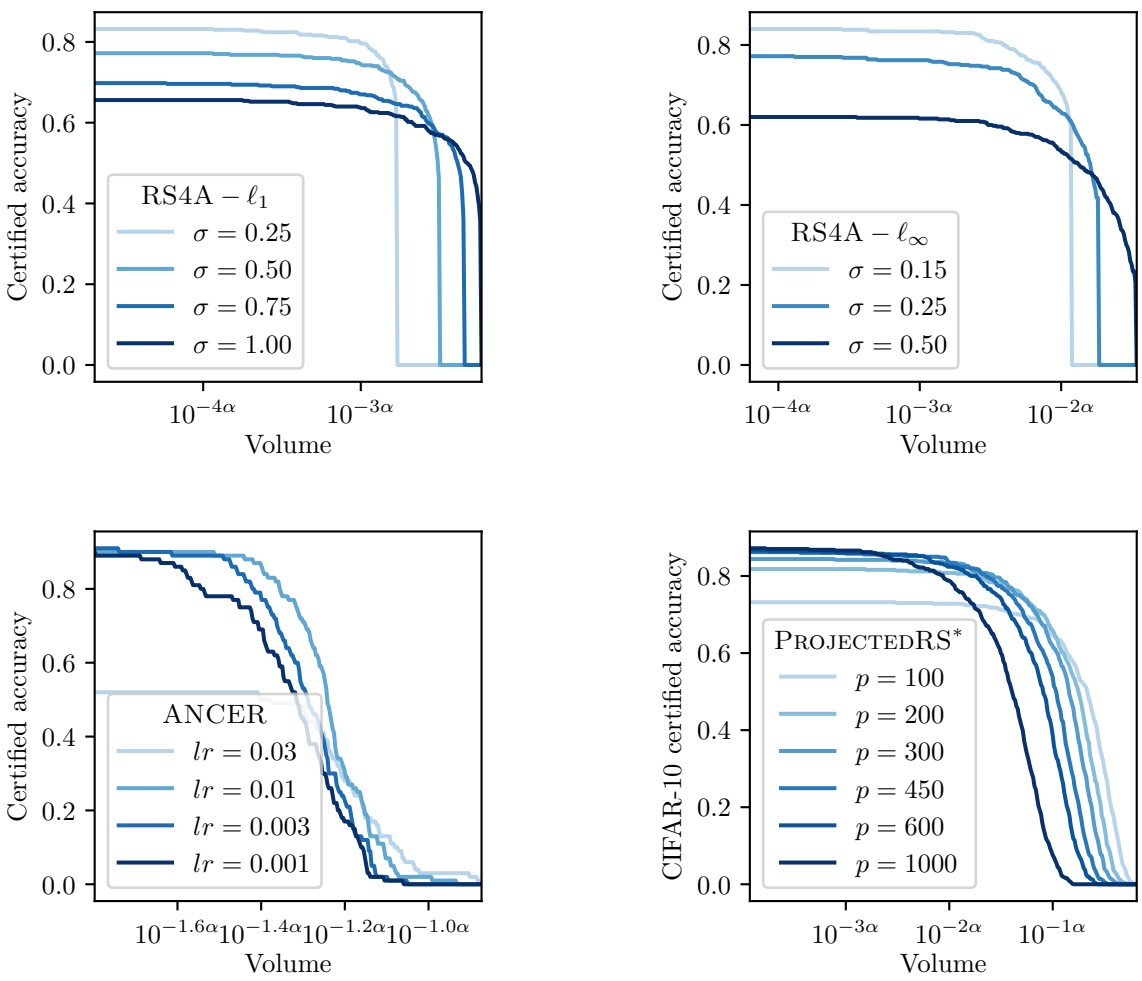

Figure 7: Hyperparameter sweeps for the CIFAR-10 dataset. Here $\alpha \approx 3465$ is a scaling constant corresponding to the $d$-dimensional unit ball volume; i.e. $V_d(B_d(1)) = 10^{-\alpha}$.

### B.5.3 SVHN certification hyperparameters

The SVHN hyperparameter sweep, shown in Figure 8, is similar to that for CIFAR-10, besides the use of fewer principal components in the projected randomized smoothing sweeps due to the higher compressibility

of the data. Our final plots in Figure 3c use $\sigma = 0.25$ for the $\ell_1$-baseline, $\sigma = 0.15$ for the $\ell_\infty$-baseline, an ANCER learning rate of 0.01, and $p = 150$ for projected randomized smoothing.

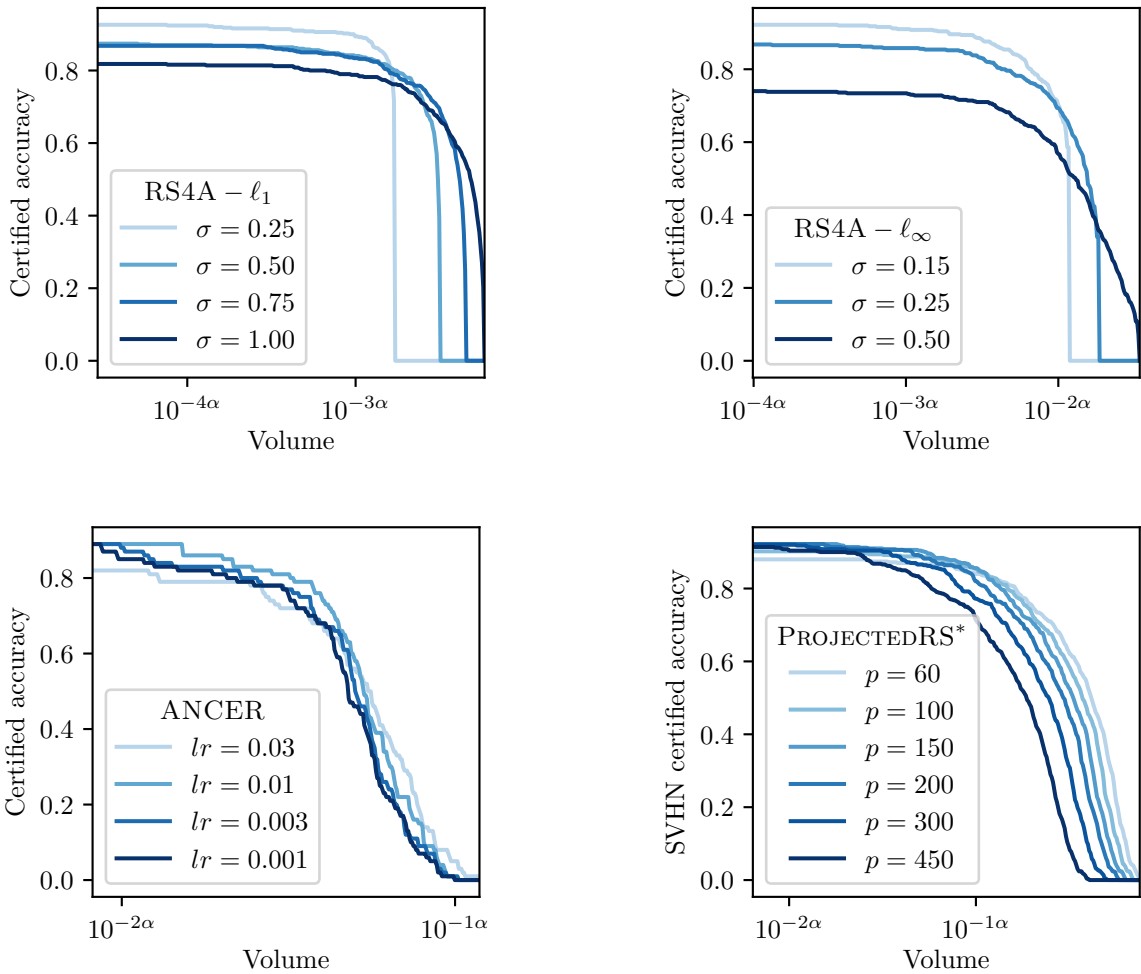

Figure 8: Hyperparameter sweeps for the SVHN dataset. Here $\alpha \approx 3465$ is a scaling constant corresponding to the $d$-dimensional unit ball volume; i.e. $V_d(B_d(1)) = 10^{-\alpha}$.

## B.6 Licenses

The CIFAR-10 dataset is covered by the MIT license, and the SVHN dataset is covered by the GPL 3 license.

