# OpenReview forum: "Projected Randomized Smoothing for Certified Adversarial Robustness"
_TMLR — Accepted by TMLR_

### Review · Reviewer_UAzk · 2023-06-14

**Summary Of Contributions:**

This paper proposes to first project the input into a low-dimensional linear subspace, then add noises in such linear subspace with randomized smoothing, and finally project back to feed into the classifier. In this way, perturbations in the null space of the projector will not change model prediction, so we can achieve a drastic improvement of certifiably robust volume. A certified volume lower bound is derived, and compared to other RS regime to show superior certified volume on CIFAR-10 and SVHN.

**Audience:**

Yes

**Broader Impact Concerns:**

No additional broader impact concerns are raised.

**Claims And Evidence:**

Yes

**Requested Changes:**

1. Large-scale experiments on ImageNet would be great.
2. Could you clarify "dominating any exponentially-scaled concessions in the numerator of (6)." a bit more on Page 7, Section 3.3?
3. Some relevant work on improving randomized smoothing certificates is not discussed, e.g., [1][2].

[1] Levine, A., and Feizi, S. Improved, deterministic smoothing for L_1 certified robustness. International Conference on Machine Learning. PMLR, 2021.
[2] Li, L., Zhang, J., Xie, T. and Li, B. Double Sampling Randomized Smoothing. International Conference on Machine Learning. PMLR, 2022.

**Strengths And Weaknesses:**

Strengths:
1. The idea of projecting to low-dimensional subspace is novel, reasonable, and theoretically sound, in terms of improving the volume of the certified region.
2. A computable lower bound of volume is proposed, which extends the existing theoretical results to bring practical value.
3. Experiments on CIFAR-10 and SVHN well justified the proposed method and also show such projection brings more robustness under existing attacks.

Weaknesses:
1. I don't think improving the volume of the certified region is a very important goal to pursue. Considering that the attacker aims to find an imperceptible adversarial example, we should not limit its ability. Hence, it should not be counted robust if the model is robust towards some perturbation directions but vulnerable to other perturbation directions, but this is indeed the goal implied by projection for maximizing certified volume.
2. Large-scale experiments on ImageNet are preferred. One unique benefit of RS is its support of large-scale datasets like ImageNet. Also, ImageNet is a challenging dataset with large intrinsic data dimensionality. Hence, to evaluate the practicality of the proposed method, ImageNet experiments would be very useful.

---

> ### Author Response · Authors · 2023-08-10
>
> Thank you for your insightful comments and suggestions. Please find our response to each raised point below.
>
> *Rebuttal*
>
> 1. "I don't think improving the volume of the certified region is a very important goal to pursue..."
>
> In contrast to your concern, our work actually certifies robustness against adversaries with a higher attack capability compared to much of the prior literature. That is, most prior works consider threat models where the adversarial attack is constrained to an isotropic ball of the form $\\{\delta : \\| \delta \\| \le R\\}$. Such works inherently assume that the adversary has limited attack ability in all directions of space. On the other hand, our method is able to handle this norm-ball threat model (Proposition 4 in the appendix) as well as adversaries that attack the system along low-variance principal components of the data distribution, even with unbounded magnitude. With our model, we are able to certify the robustness against such attacks, whereas prior isotropic randomized smoothing-based methods cannot. Thus, we are not making the model vulnerable to certain perturbation directions, but actually the exact opposite: we are adding robustness along other directions. The directions we robustify against are the low-variance principal component directions. We show in Section 4.1 that attacks along these directions pose a viable threat.
>
> We emphasize that there is significant interest in extending certified robustness to anisotropic regions, since in general this allows us to certify larger and more general regions of the input space [1-4]; we believe this is of particular interest to the TMLR community as [1] was published in this very journal. In all of these works, the natural scalar measure of size of the certified region is indeed taken to be volume, since there is not a uniform "radius" in such anisotropic cases. In addition to our volume lower bound, we characterize the geometry of our anisotropic certified region (see Propositions 1 and 2).
>
> [1] Eiras, Francisco, et al. "Ancer: Anisotropic certification via sample-wise volume maximization." Transactions on Machine Learning Research. 2022.
>
> [2] Liu, Chen, Ryota Tomioka, and Volkan Cevher. "On certifying non-uniform bounds against adversarial attacks." International Conference on Machine Learning. PMLR, 2019.
>
> [3] Rumezhak, Taras, et al. "RANCER: Non-Axis Aligned Anisotropic Certification with Randomized Smoothing." Proceedings of the IEEE/CVF Winter Conference on Applications of Computer Vision. 2023.
>
> [4] Tecot, Lucas Matthew. Robustness verification with non-uniform randomized smoothing. University of California, Los Angeles, 2021.
>
> 2. "Large-scale experiments on ImageNet are preferred.."
>
> We appreciate your and Reviewer xS7n's interest regarding applying our method to ImageNet. The primary challenge to executing our method for higher-dimensional datasets is solving the $\ell_{\infty}$ norm minimization (Alg1). For larger images, holding the entire problem in memory is not directly feasible on our hardware. While some methods have been proposed regarding these large-scale $\ell_{\infty}$ regression problems (e.g. Shen 2014 in the paper bibliography), we consider these extensions outside the scope of our work. We have updated the draft to make this point more clear.
>
> *Requested Changes*
>
> 1. "Large-scale experiments on ImageNet would be great."
>
> Please see rebuttal 2 above. We have expanded on the limitations discussion in Section 3.4 in the revised manuscript.
>
> 2. "Could you clarify "dominating any exponentially-scaled concessions in the numerator of (6)." a bit more on Page 7, Section 3.3?"
>
> We have added the following clarifying text to Section 3.3:
>
> "while the numerator in (6) might be *exponentially* smaller than that of (8), the denominator is smaller by a *factorial* factor. We thus expect the volumes of projected randomized smoothing to dominate at higher dimensions."
>
> 3. "Some relevant work on improving randomized smoothing certificates is not discussed, e.g., [1][2]."
>
> Thank you for bringing this to our attention. We have discussed these relevant works in our revised manuscript.

---

> > ### Comment · Reviewer_UAzk · 2023-08-15
> > **Thanks for clarification, and some concerns remain.**
> >
> > > 1. "I don't think improving the volume of the certified region is a very important goal to pursue..."
> > In contrast to your concern, our work actually certifies robustness against adversaries with a higher attack capability compared to much of the prior literature. That is, most prior works consider threat models where the adversarial attack is constrained to an isotropic ball of the form $\{\delta: ||\delta|| \le R\}$. Such works inherently assume that the adversary has limited attack ability in all directions of space. On the other hand, our method is able to handle this norm-ball threat model (Proposition 4 in the appendix) as well as adversaries that attack the system along low-variance principal components of the data distribution, even with unbounded magnitude.
> >
> > I respectfully disagree with *our work actually certifies robustness against adversaries with a higher attack capability compared to much of the prior literature*. Proposition 4 only shows that when the proposed method has radius $R$, it implies an all-round norm $R$ radius ball. However, it would likely be the case that, without the proposed method for 80% input we can have projected robust radius $R$; with the proposed method only for 20% input, we can have robust radius $R$.
> >
> > Other concerns are well addressed and explained. Thanks! Though the scalability barrier incurred by $\ell_\infty$ norm optimization poses a strong scalability limitation compared to randomized smoothing, which should be noted for this work.

---

> > > ### Author Response · Authors · 2023-08-15
> > >
> > > Thank you for your updated comments. Your remaining concern poses an important question to consider--however, we show experimentally in Appendix B.3 that this does not occur. Figure 6 demonstrates that the certified radii curves for randomized smoothing and projected randomized are virtually identical. Therefore, our certificates generally contain a norm ball of comparable size to standard randomized smoothing in addition to the span of low-variance components.
> > >
> > > We have expanded on the discussion of the $\ell_{\infty}$ optimization problem in Section 3.4 in the newest revision of the manuscript.
> > >
> > > Please let us know if you have any remaining concerns or questions on our work.

---

> > > > ### Comment · Reviewer_UAzk · 2023-08-19
> > > >
> > > > Thanks for the reply. From Appendix B.3 and Figure 6, I don't fully agree that `the certified radii curves for randomized smoothing and projected randomized are virtually identical` - the projected RS curve is very slightly --- but always --- lower than the curves of randomized smoothing, which means that `our work actually certifies robustness against adversaries with a higher attack capability compared to much of the prior literature` is not absolutely true. For isotropic attackers, the framework provides a slightly inferior robustness, as the trade-off for much robustness against anisotropic attackers that exert perturbations on certain dimensions.

---

> > > > > ### Author Response · Authors · 2023-08-19
> > > > >
> > > > > We appreciate the reviewer's interest in comparing adversary strengths. Our perspective is that we are able to certify against adversaries which have nearly the same isotropic radius but also the entire low-variance subspace to attack with. As the reviewer pointed out, the isotropic radius difference in Figure 6 is very small, while the added perturbation possibilities along the low-variance subspace are substantial. This claim is made rigorous when we use volume as a measure of "strength", which is standard when considering anisotropic adversaries; we see that indeed our certified volumes are much larger than RS.

---

### Review · Reviewer_xS7n · 2023-06-29

**Summary Of Contributions:**

The paper presents a randomized smoothing (RS) technique that improves the volume of the certified region, by using dimension-reducing projection. The paper computes a theoretical lower bound on the volume of the certified region. The evaluation shows that on CIFAR10 and SVHN, the technique yields larger volumes than prior SOTA methods.

**Audience:**

Yes

**Claims And Evidence:**

Yes

**Requested Changes:**

1. Provide an ablation study showing the effectiveness of ProjectedRS using PCA dimensions over any randomly picked dimensions.

2. Add more explanation on why validating the effectiveness of the technique on ImageNet is hard and not done in this work

**Strengths And Weaknesses:**

**Strengths:**
1. The paper presents a novel projected RS approach
2. The theoretical analysis shows that this approach in theory results in improved lower bound on the certified volume
3. The paper is well-written and easy to follow

**Weaknesses:**
1. There is no ablation study to show that projecting on the Principal Component (PCA) works better than even projecting on random dimensions in terms volume of the certified region.

2. This work does not show the effectiveness of the proposed technique on ImageNet. The explanation of why it is hard to validate its usefulness for ImageNet is quite short. It just says "would likely need to leverage techniques from the large-scale l_\inf-regression literature". This explanation is quite unclear.

**Questions:**
1. Does the theoretical lower bound on the volume hold irrespective of the use of principal components? If yes, then can you explicitly state it in the text

---

> ### Author Response · Authors · 2023-08-10
>
> Thank you for your constructive suggestions. Please find our responses to each raised point below.
>
> *Rebuttal*
>
> 1. "There is no ablation study to show that projecting on the Principal Component (PCA) works better than even projecting on random dimensions..."
>
> We have included the relevant experiment in Appendix B.1 of the revised manuscript. Projecting onto random subspaces necessitates a higher number of components to get the same clean accuracy. This causes a substantial drop in certification performance compared to PCA projections, as we document in the appendix.
>
> 2. "This work does not show the effectiveness of the proposed technique on ImageNet..."
>
> We appreciate your and Reviewer UAzk's interest regarding applying our method to ImageNet. The primary challenge to executing our method for higher-dimensional datasets is solving the $\ell_{\infty}$ norm minimization (Alg1). For larger images, holding the entire problem in memory is not directly feasible on our hardware. While some methods have been proposed regarding these large-scale $\ell_{\infty}$ regression problems (e.g. Shen 2014 in the paper bibliography), we consider these extensions outside the scope of our work. We have updated the draft to make this point more clear.
>
> *Questions*
>
> 1. "Does the theoretical lower bound on the volume hold irrespective of the use of principal components? If yes, then can you explicitly state it in the text"
>
> Yes, the lower bounds given by Theorem 2 and Proposition 3 hold for arbitrary projections $P \colon x \mapsto U^\top x$ ($U$ does not need to be composed of principal components). This makes our theoretical robustness certificate general and applicable to projected randomized smoothing using non-PCA-based linear projection methods. Of course, choosing the projection matrix $U^\top$ intelligently will result in a higher certified radius $R$ in the lower-dimensional space, making the overall certified volume lower bound given by (6) and (7) larger. We see the effects of this in our newly added ablation study comparing our certificates using PCA-based projection versus random projection---PCA yields notably better performance. We have updated the manuscript to clarify these aspects.
>
> *Requested Changes*
>
> 1. "Provide an ablation study showing the effectiveness of ProjectedRS using PCA dimensions over any randomly picked dimensions."
>
> Please see rebuttal 1 above. We have revised the text to include an ablation test against randomly picked directions.
>
> 2. "Add more explanation on why validating the effectiveness of the technique on ImageNet is hard and not done in this work"
>
> Please see rebuttal 2 above. We have expanded on the limitations discussion in Section 3.4 in the revised manuscript.

---

> > ### Comment · Reviewer_xS7n · 2023-08-20
> >
> > Thanks to the authors for providing clarification and making the requested changes.

---

### Review · Reviewer_65nC · 2023-08-07

**Summary Of Contributions:**

This paper considers the technique of certified adversarial robustness where the input data is first projected down to a smaller dimension and then randomized smoothing is applied on the dimension-reduced input. For this technique, the paper proposes new lower bounds on the volume of the certified region. Experimentally, the paper shows that the directions along which the dimensions were reduced could have led to adversarial vulnerabilities, and that the volume of the certified region using this technique is larger than what is given by existing techniques.

**Audience:**

Yes

**Broader Impact Concerns:**

No concerns of ethical implications.

**Claims And Evidence:**

Yes

**Requested Changes:**

Apart from the weaknesses in the section above, I have the following suggestion: The authors can include a discussion on if/how the results could be extended to norms other than the $l_2$ norm.



**Strengths And Weaknesses:**

Strengths:
1. The paper is the first to prove lower bound on the volume for the certified region when applying randomized smoothing on reduced dimensionality.
2. The experiments demonstrates experimentally that adversarial vulnerabilities exist along subspaces which are not important for classification.
3. Experimentally, the paper confirms that the volume certified by this technique is larger than those of existing techniques.

Weaknesses:
1. Volume might not be the best metric for certified adversarial robustness. For example, the standard 'minimum norm attack' metric (under various norms) is more suitable. Volume can be large, and yet there could be a small norm successful attack possible.

2. The experiments in Section 4.1 consider $l_{\infty}$ attacks, but the theory is proved for $l_2$ attacks.

3. The theoretical results are somewhat intuitive and the proofs seem relatively straight-forward.

---

> ### Author Response · Authors · 2023-08-10
>
> Thank you for your comments and helpful suggestions. We have addressed them below, and revised the manuscript accordingly.
>
> *Rebuttal*
>
> 1. "Volume might not be the best metric for certified adversarial robustness...."
>
> We note that our certificates can be considered an improvement over a typical norm-ball certificate. By Proposition 4 in the appendix, our certificates in fact contain typical $\ell_2$ norm ball as a subset. We additionally expand this certificate by capturing the space of low-variance directions in the dataset. While finding a common metric for these different certificate geometries is challenging, Lebesgue measure / volume is fairly well established in the literature and we believe it to be adequate for the task at hand.
>
> We emphasize that there is significant interest in extending certified robustness to anisotropic regions, since in general this allows us to certify larger and more general regions of the input space [1-4]; we believe this is of particular interest to the TMLR community as [1] was published in this very journal. In all of these works, the natural scalar measure of size of the certified region is indeed taken to be volume, since there is not a uniform "radius" in such anisotropic cases. In addition to our volume lower bound, we characterize the geometry of our anisotropic certified region (see Propositions 1 and 2).
>
> [1] Eiras, Francisco, et al. "Ancer: Anisotropic certification via sample-wise volume maximization." Transactions on Machine Learning Research. 2022.
>
> [2] Liu, Chen, Ryota Tomioka, and Volkan Cevher. "On certifying non-uniform bounds against adversarial attacks." International Conference on Machine Learning. PMLR, 2019.
>
> [3] Rumezhak, Taras, et al. "RANCER: Non-Axis Aligned Anisotropic Certification with Randomized Smoothing." Proceedings of the IEEE/CVF Winter Conference on Applications of Computer Vision. 2023.
>
> [4] Tecot, Lucas Matthew. Robustness verification with non-uniform randomized smoothing. University of California, Los Angeles, 2021.
>
> 2. "The experiments in Section 4.1 consider $l_{\infty}$ attacks, but the theory is proved for $l_2$ attacks."
>
> The certified region of our classifier consists of a low-dimensional $\ell_2$ ball "extruded" along the projection null space. The purpose of the experiments in Section 4.1 is to establish that this low-variance subspace indeed contains adversarial perturbations. We construct such adversarial attacks by restricting an $\ell_{\infty}$ attack to the subspace. As Reviewer 65nC hints at, this experiment could also have been conducted by restricting other reasonable norm-ball attacks to the subspace; these would all provide evidence that the low-variance subspace contains adversarial perturbations. As our architecture would certifiably defend against all such attacks, we see the choice of adversary in Section 4.1 as independent of the choice of smoothing norm in the low-dimensional subspace. We simply chose $\ell_{\infty}$ attacks because they are the most standard in the literature.
>
> The related question of using different smoothing methods, designed for different norms, in the low-dimensional space is a highly interesting one. We discuss it below in Change 1.
>
> 3. "The theoretical results are somewhat intuitive and the proofs seem relatively straight-forward."
>
> We feel that our theory leverages lesser-known results from the mathematical literature (Theorem 1) in a rather elegant way. We take pride in you finding the theory easy to follow; in our opinion, that not only validates the effort we've placed in clear mathematical presentation but also suggests that our results have intuitive interpretations that ideally most readers will be able to follow.
>
>
> *Requested Changes*
>
> 1. "Include a discussion on if/how the results could be extended to norms other than the $l_2$ norm."
>
> Reviewer 65nC raises the interesting question of whether other smoothing norms could be used in the low-dimensional projected space. For instance, could we certify an $\ell_1$-ball in the projected space using [1] and then extrude it along the null space? Propositions 1 and 2 in the paper would hold as is, with norms referring to the $\ell_1$-norm. We believe Theorem 2 also holds, as the relevant observation in the proof, $\| y_1 \|_{\infty} \leq \| y_1 \|$, holds for any p-norm on the right hand side. Such extensions are an interesting area of future work, and we have updated the manuscript to discuss these possibilities further.
>
> [1] Levine, A., and Feizi, S. Improved, deterministic smoothing for L_1 certified robustness. International Conference on Machine Learning. PMLR, 2021.

---

> > ### Comment · Reviewer_65nC · 2023-08-28
> > **Response to the authors**
> >
> > Thank you for addressing my concerns and updating the manuscript with the requested changes. I am still not fully convinced about the practical usefulness of just certifying the volume. However the technique proposed also certifies a minimum l2 radius (which was also discussed between Reviewer UAzk and the authors). Hence this particular technique would not have the particular problem of certifying a large volume while having a small certified radius.

---

### Author Response · Authors · 2023-08-10

We sincerely thank the reviewers for their insightful comments and valuable suggestions. Please see our individual responses to each reviewer, where we address each point raised and mention the corresponding revisions we have made to the manuscript. Revisions to the manuscript are highlighted in blue.

---

### Author Response · Authors · 2023-08-19

We thank all reviewers again for their time taken in evaluating our manuscript. As the discussion period closes in two days, we would like to remind reviewers that we are happy to discuss any remaining questions that they may have about our work.

---

### Decision · Action_Editors · 2023-09-11

**Recommendation:** Accept as is

**Comment:**

The paper suggests an alternative robustness metric, where adversarial perturbations are constrained to lie on a data manifold chosen as a subspace coming out of a principal components analysis. The authors show that performing randomized smoothing in this restricted subspace leads to stronger guarantees (wrt to the restricted threat model). The reviewers raised questions on the notion of robustness considered and the value of the PCA (vs random projections) that were adequately addressed during the discussion phase. Hence I recommend acceptance.

**Audience:**

The problem of developing classifiers that are robust to adversarial perturbations is a deep and challenging one, and despite significant progress in the area, finding highly performant classifiers that are certifiably robust to worst case perturbations, even under limited threat models like linf norm constrained perturbations, remains open. The authors suggest a promising path forward by identifying a subspace that restricts adversarial perturbations to live on a data manifold, and show that this enables significant improvements in the accuracy robustness tradeoff. This should be of broader interest to the deep learning community, particularly those focused on security and robustness challenges in deep learning.

**Claims And Evidence:**

The authors develop a variant of randomized smoothing that involves projecting onto a subspace that ideally captures the data manifold, and show that by only seeking robustness within this subspace, they can improve results on certified robustness of classifiers to adversarial perturbations on the data manifold. The authors present clear theoretical and empirical justifications for their claims and all reviewers were in consensus about this.

---

> ### Author Response · Authors · 2023-09-25
> **Thank you, and camera-ready uploaded.**
>
> Dear AE and Reviewers,
>
> Thank you all for your constructive feedback and insightful discussions! We are very excited to have our paper published by TMLR.
>
> We have uploaded the camera-ready version of the manuscript, as well as a link to our code.